# KineFlow: Kinematic Second-Order Flow Matching for Time-Series Forecasting

**Haiqi Jiang** [1]    **Hui Xiong** [1 2]

## Abstract

Conventional time-series discriminative forecasting relies on point-wise regression, which inherently induces over-smoothing and fails to capture stochastic volatility in complex systems. While first-order generative flow matching methods mitigate this issue, they ignore system inertia, resulting in phase-space ambiguities and high sensitivity to noise. We introduce KineFlow, a generative time-series forecasting framework that augments flow matching with a phase-space Neural Acceleration Field, treating exogenous inputs as driving forces that produce gradual momentum shifts rather than abrupt state perturbations. This second-order formulation serves as a structural filter via double integration, suppressing high-frequency noise and producing robust, physically consistent predictions. Extensive experiments on six real-world benchmarks demonstrate that Kine-Flow achieves an average 15% MSE improvement over discriminative baselines and an 8% gain in CRPS compared to state-of-the-art generative methods.

## 1. Introduction

Time-series forecasting is essential for modeling complex dynamical systems, from volatile energy loads to extreme weather events (Gasparin et al., 2021; Fang et al., 2021). Recent time-series forecasting has focused on discriminative deep learning, ranging from specialized Transformers to large-scale foundation models (Wen et al., 2023; Liang et al., 2024). While these methods excel at capturing global dependencies (Zhou et al., 2021), their reliance on point-wise objectives (e.g., MSE) tends to produce unimodal forecasts and thus underrepresent predictive uncertainty (Barnett

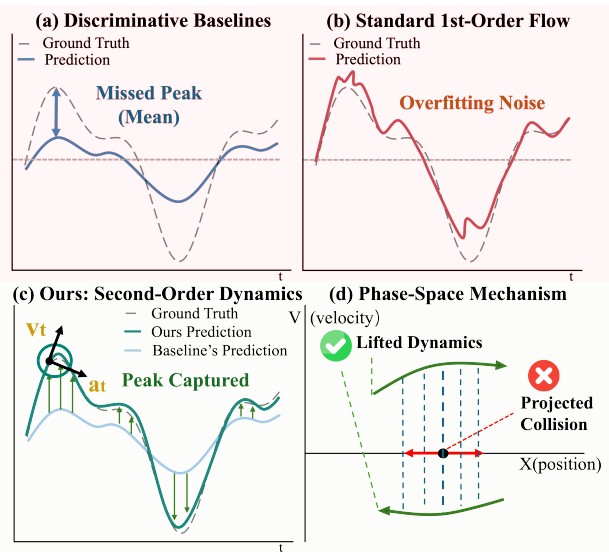

*Figure 1.* Comparison of forecasting paradigms. (a) Discriminative models suffer from over-smoothing. (b) First-order flows exhibit noise overfitting. (c) KineFlow recovers inertia via second-order dynamics ($a_t$). (d) Phase-space lifting resolves conflicts.

et al., 2004; Mathieu et al., 2015; Rasul et al., 2021). As illustrated in Fig. 1(a), conditional-mean predictions result in over-smoothing, suppressing high-frequency volatility.

To overcome the limitations of discriminative point forecasting, recent work has increasingly turned to generative modeling (Brophy et al., 2023), which models the full conditional predictive distribution to capture detailed and stochastic dynamics. While diffusion models (Yang et al., 2023) excel at modeling complex distributions, their stochastic denoising processes lead to slow convergence and high inference latency. In contrast, Flow Matching (FM) constructs a deterministic Ordinary Differential Equation (ODE)-based probability path, enabling efficient and physically interpretable temporal modeling (Lipman et al., 2023).

However, standard Conditional Flow Matching (CFM) typically defines the vector field as the instantaneous velocity. Although straightforward, the first-order formulation suffers from both numerical and representational limitations. Numerically, high-frequency errors in the network output are weakly attenuated by single integration, making first-order flows prone to noise amplification (Fig. 1(b)). Rep-

[1]Thrust of Artificial Intelligence, The Hong Kong University of Science and Technology (Guangzhou) [2]China Department of Computer Science and Engineering, The Hong Kong University of Science and Technology Hong Kong SAR, China. Correspondence to: Hui Xiong <xionghui@ust.hk>.

*Proceedings of the 43$^{rd}$ International Conference on Machine Learning*, Seoul, South Korea. PMLR 306, 2026. Copyright 2026 by the author(s).

resentationally, when observations are treated as states, a first-order continuous vector field cannot model intersecting trajectories (Fig. 1(d)) due to the topological constraints of ODE flows (Dupont et al., 2019). In time-series forecasting, identical past states can lead to multiple future trajectories because of stochasticity or external factors, but first-order FM enforces a single velocity at each position, producing averaged and physically implausible predictions.

In this work, we propose KineFlow, a framework that introduces a second-order kinematic inductive bias into flow matching. We lift the generative process to phase space (position and velocity) by treating each observed system state as a particle, and model a Neural Acceleration Field that predicts instantaneous acceleration. This second-order formulation serves as a structural low-pass filter via double integration, attenuating high-frequency noise and yielding physically consistent, expressive temporal dynamics.

Mechanically, KineFlow establishes a hierarchical kinematic framework. First, a deterministic backbone encodes historical observations to construct an informative kinematic prior, providing the system with initial position and inertial momentum. Second, we introduce a force-based conditioning mechanism for exogenous variables (e.g., humidity for electricity demand). Instead of treating them as generic context, we model external factors as driving forces within the acceleration field, producing gradual momentum shifts rather than direct state perturbations. Finally, by evolving these components via a second-order ODE solver, the system leverages physical inertia to buffer against input noise, maintaining physically consistent dynamics while naturally filtering artifacts (Fig. 1(c)).

We evaluate KineFlow on six real-world datasets, achieving state-of-the-art results. Quantitatively, KineFlow outperforms discriminative methods by 15% in MSE and surpasses leading generative models by 8% in CRPS. We provide theoretical analysis confirming that our second-order formulation effectively bounds high-frequency error propagation. Furthermore, our visualizations highlight KineFlow's interpretable, physically consistent trajectories, resolving phase-space ambiguities and capturing dynamic variability.

Our contributions are summarized as follows:

- We propose a phase-space generative model KineFlow that resolves trajectory ambiguities and reduces high-frequency noise via second-order integration.

- We develop a Neural Acceleration Field with force-based conditioning, treating exogenous factors as driving forces to ensure kinematic consistency.

- Extensive evaluation on six benchmarks demonstrates that KineFlow outperforms both discriminative and generative baselines, achieving a favorable balance of accuracy and probabilistic calibration.

## 2. Related Work

**Time Series Forecasting Methods**. Transformer-based models dominate time-series forecasting (Wen et al., 2023), with representative works such as Autoformer (Wu et al., 2021) and DLinear (Zeng et al., 2023) advancing trend-season decomposition and strong linear baselines, respectively. Later models such as PatchTST (Nie et al., 2023), iTransformer (Liu et al., 2024), and TimeXer (Wang et al., 2024b) further enhance local representations, inter-variable modeling, and the fusion of endogenous and exogenous information. Large pre-trained foundation models for time series (Liang et al., 2024; Liu et al., 2025b), such as Moirai (Woo et al., 2024), show strong zero-shot generalization. Despite their strength, these models rely on pointwise objectives, producing over-smoothed, unimodal forecasts.

**Generative Model for Time Series Generation**. Generative approaches for time-series forecasting (Yang et al., 2023) model the full conditional distribution to capture stochastic uncertainty. Diffusion-based methods such as TimeGrad (Rasul et al., 2021), CSDI (Tashiro et al., 2021), and TimeDiff (Kollovieh et al., 2023) improve conditioning and multi-scale coherence, while Diffusion-TS (Yuan & Qiao, 2024) handles multiple frequency components. More recently, CMA (Jiang et al., 2025) combines diffusion and meta-learning to handle varying historical and prediction horizons without retraining. Despite their expressiveness, diffusion models require iterative sampling. Flow Matching (FM) (Lipman et al., 2023; Tong et al., 2024) frames continuous flows as an optimal transport problem to straighten trajectories and improve efficiency. TSFlow (Kollovieh et al., 2025) applies FM for faster, more accurate forecasts. Standard FM, however, is first-order and neglects inertia and exogenous effects, motivating second-order extensions for smoother, momentum-aware dynamics.

**Second-Order Dynamics in Time-Series Modeling**. Many real-world temporal systems exhibit momentum-driven dynamics. Second-order continuous models, including augmented Neural ODEs (Chen et al., 2018; Dupont et al., 2019; Norcliffe et al., 2020), and latent position-momentum models (Yildiz et al., 2019), demonstrate that incorporating acceleration enhances representational power for complex sequences. While recent high-order flow matching extensions provide statistical guarantees (Cao et al., 2025; Chen et al., 2025a; Su et al., 2025; Chen et al., 2025b), their lack of strict physical coupling between derivatives leaves them vulnerable to integration drift. Inspired by these works, we introduce a second-order generative framework that captures kinematically continuous, momentum-driven time-series dynamics.

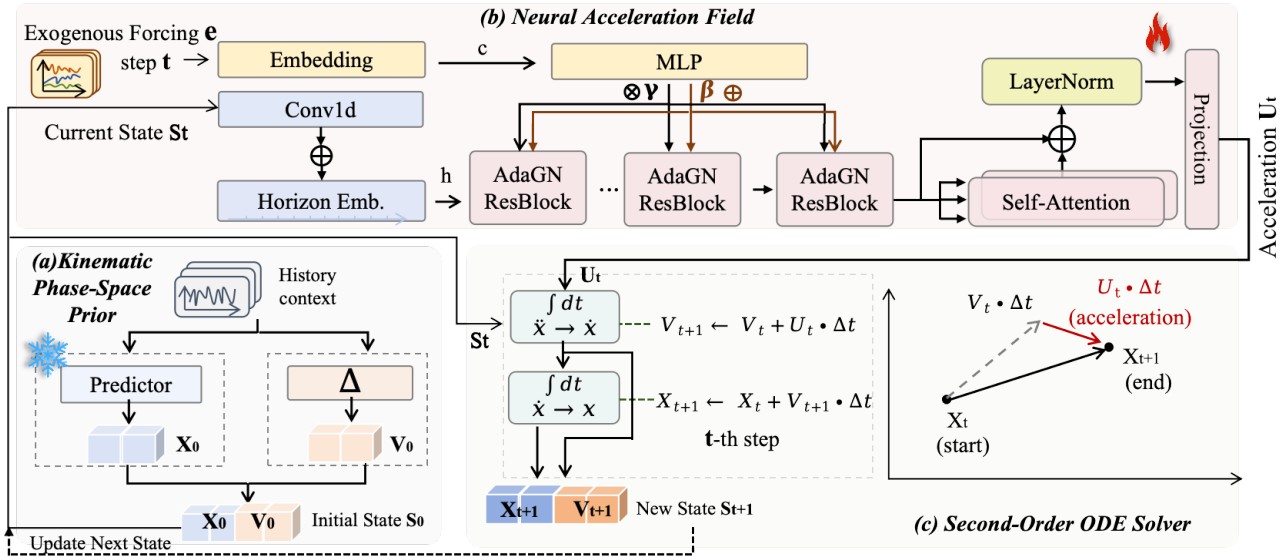

*Figure 2.* Framework of KineFlow. (a) Kinematic Phase-Space Prior: combines a deterministic predictor and finite-difference estimation to initialize $\mathbf{S}_0 = [\mathbf{X}_0, \mathbf{V}_0]$. (b) Neural Acceleration Field: predicts acceleration with horizon embeddings and AdaGN ResBlocks to incorporate exogenous forcing. (c) Second-Order ODE Solver: integrates acceleration to update the phase-space state.

## 3. Preliminaries

**Problem Definition.** We consider a multivariate time series $\boldsymbol{x} \in \mathbb{R}^{T \times D}$ with $T$ time steps and $D$ features:

$$\boldsymbol{x} = [\boldsymbol{x}^H, \boldsymbol{x}^F] = [\underbrace{x_1, \ldots, x_{T_H}}_{\text{observed history}}, \underbrace{x_{T_H+1}, \ldots, x_{T_H+T_F}}_{\text{future target}}],$$

(1)

where $\boldsymbol{x}^H$ denotes the observed historical context of length $T_H$ and $\boldsymbol{x}^F$ denotes the future target of length $T_F$. We aim to model the conditional predictive distribution $p_\theta(\boldsymbol{x}^F \mid \boldsymbol{x}^H)$ of future, capturing the uncertainty in temporal evolution.

**Conditional Flow Matching (CFM).** CFM ([Lipman et al., 2023](); [Tong et al., 2024]()) generates data via the ODE $d\mathbf{z}_t/dt = v_\theta(\mathbf{z}_t, t)$, where $\mathbf{z}_0$ and $\mathbf{z}_1$ are source and target samples, and $v_\theta$ predicts the flow velocity. The network is trained to match the linear Optimal Transport (OT) path $\mathbf{z}_t = (1 - t)\mathbf{z}_0 + t\mathbf{z}_1$, corresponding to a constant target velocity $u_t = \mathbf{z}_1 - \mathbf{z}_0$. The objective is

$$\mathcal{L}_{\text{CFM}}(\theta) = \mathbb{E}_{t,\mathbf{z}_0,\mathbf{z}_1} \left[ \|v_\theta(\mathbf{z}_t, t) - u_t\|^2 \right].$$

(2)

While efficient, the linear OT path enforces zero acceleration ($\ddot{\mathbf{z}}_t = 0$) and thus ignores momentum-driven dynamics present in real-world time series.

## 4. Methodology

In this work, we introduce KineFlow, a time-series generative forecasting method based on second-order phase-space dynamics. As illustrated in Fig. 2, KineFlow is structured around three components: (a) a *Kinematic Phase-Space Prior*, which initializes position and velocity from historical

observations to encode inertial trends; (b) a *Neural Acceleration Field*, which models the driving forces conditioned on the evolving phase-space state and exogenous inputs; and (c) a *Second-Order ODE Solver*, which propagates the system forward in time. This decomposition separates forecasting: the prior captures base motion, the acceleration field models nonlinear dynamics, and the solver enforces kinematic consistency.

### 4.1. Second-Order Phase-Space Formulation

To capture continuous time-series dynamics, we lift the generative process to a high-dimensional phase space and model the flow over a continuous integration time $t \in [0, 1]$, evolving the system from the prior state to the target future sequence. We formulate the system in phase space by treating the time-series values as position coordinates.

**Phase-Space State Definition.** Let $\mathbf{X}_t \in \mathbb{R}^{T_F \times D}$ denote the generated positions at integration time $t$. Unlike first-order models which evolve only positions, we explicitly introduce a velocity variable $\mathbf{V}_t = \frac{d}{dt}\mathbf{X}_t \in \mathbb{R}^{T_F \times D}$ to represent the momentum of the flow. The complete system state is the concatenation of position and velocity:

$$\mathbf{S}_t = [\mathbf{X}_t, \mathbf{V}_t]^\top \in \mathbb{R}^{T_F \times 2D}.$$

(3)

**Generative Dynamics.** We model the generative flow in phase space using a second-order ODE, parametrized by a Neural Acceleration Field $\mathcal{A}_\theta$:

$$\frac{d}{dt} \begin{bmatrix} \mathbf{X}_t \\ \mathbf{V}_t \end{bmatrix} = \begin{bmatrix} \mathbf{V}_t \\ \mathcal{A}_\theta(\mathbf{S}_t, t, \mathbf{e}) \end{bmatrix},$$

(4)

where $\mathbf{e}$ represents exogenous forcing. The acceleration $\mathbf{U}_t = \mathcal{A}_\theta(\mathbf{S}_t, t, \mathbf{e}) \in \mathbb{R}^{T_F \times D}$ governs the rate of change of velocity. By modeling the dynamics at the acceleration level, the trajectory $\mathbf{X}_t$ is obtained through double integration, which explicitly incorporates kinematic inertia and attenuates high-frequency noise in the generative flow.

### 4.2. Kinematic Phase-Space Prior

As illustrated in Fig. 2(a), we define the initial state $\mathbf{S}_0 = [\mathbf{X}_0, \mathbf{V}_0]^\top$ for the generative flow in Eq. 4 using an informative phase-space prior derived from the historical context:

$$\mathbf{S}_0 \sim \mathcal{N}(\boldsymbol{\mu}_0, \sigma^2 \mathbf{I}), \quad \boldsymbol{\mu}_0 = [\boldsymbol{\mu}^X, \boldsymbol{\mu}^V]^\top, \qquad (5)$$

where $\boldsymbol{\mu}^X$ and $\boldsymbol{\mu}^V$ serve as the prior means for $\mathbf{X}_0$ and $\mathbf{V}_0$, respectively, and $\sigma$ governs the stochasticity of the initialization.

The prior position $\boldsymbol{\mu}^X$ is obtained from a pre-trained deterministic predictor $\mathcal{F}_\phi$ (Wu et al., 2021):

$$\boldsymbol{\mu}^X = \mathcal{F}_\phi(\boldsymbol{x}^H) \in \mathbb{R}^{T_F \times D}. \qquad (6)$$

The prior velocity $\boldsymbol{\mu}^V$ preserves kinematic continuity by repeating the last-step difference from the historical context along the forecast horizon:

$$v = x_{T_H} - x_{T_H - 1} \in \mathbb{R}^D, \quad \boldsymbol{\mu}^V = \mathbf{1}_{T_F} v^\top \in \mathbb{R}^{T_F \times D}, \qquad (7)$$

which ensures the generative flow inherits the momentum from history.

This prior naturally supports stochastic generation ($\sigma > 0$) for probabilistic forecasts and deterministic refinement ($\sigma = 0$) for point prediction.

### 4.3. Neural Acceleration Field

The Neural Acceleration Field $\mathcal{A}_\theta$ models the time-dependent force field governing the phase-space flow. $\mathcal{A}_\theta$ is implemented as a hybrid architecture that combines local kinematic consistency with global temporal coherence, producing the acceleration $\mathbf{U}_t = \mathcal{A}_\theta(\mathbf{S}_t, t, \mathbf{e})$.

The phase-space state $\mathbf{S}_t$ is projected to a $d$-dimensional latent feature space through a 1D convolutional operator acting along the forecast horizon. To preserve horizon-wise ordering, we add a learnable positional embedding $\mathbf{E}_{\text{pos}}$:

$$\mathbf{H}^{(0)} = \text{Conv1D}_\phi(\mathbf{S}_t) + \mathbf{E}_{\text{pos}} \in \mathbb{R}^{T_F \times d}. \qquad (8)$$

The latent representation is updated by a stack of $N$ residual blocks (He et al., 2016). The integration time $t$ and exogenous forcing $\mathbf{e}$ are combined into a condition vector $\mathbf{c}$, which modulates the latent evolution through Adaptive Group Normalization (AdaGN) (Dhariwal & Nichol, 2021). Specifically, for the $n$-th block,

$$\mathbf{H}^{(n+1)} = \mathbf{H}^{(n)} + \mathcal{R}\Big(\text{AdaGN}\big(\mathbf{H}^{(n)}, \mathbf{c}\big)\Big), \qquad (9)$$

**Algorithm 1** Inference via KineFlow

1: **Input:** History $\boldsymbol{x}^H$, exogenous $\mathbf{e}$, steps $K$
2: $\boldsymbol{\mu}^X \leftarrow \mathcal{F}_\phi(\boldsymbol{x}^H), \boldsymbol{\mu}^V \leftarrow \mathbf{1}_{T_F}(x_{T_H} - x_{T_H - 1})^\top$
3: $\mathbf{S}_0 \leftarrow [\boldsymbol{\mu}^X, \boldsymbol{\mu}^V]^\top + \sigma\boldsymbol{\epsilon}, \quad \boldsymbol{\epsilon} \sim \mathcal{N}(\mathbf{0}, \mathbf{I})$
4: $\Delta t \leftarrow 1/K$
5: **for** $k = 0$ **to** $K - 1$ **do**
6: $\quad \mathbf{U}_t \leftarrow \mathcal{A}_\theta(\mathbf{S}_t, k\Delta t, \mathbf{e})$
7: $\quad \mathbf{V}_{t+\Delta t} \leftarrow \mathbf{V}_t + \mathbf{U}_t \Delta t$
8: $\quad \mathbf{X}_{t+\Delta t} \leftarrow \mathbf{X}_t + \mathbf{V}_{t+\Delta t} \Delta t$
9: $\quad \mathbf{S}_{t+\Delta t} \leftarrow [\mathbf{X}_{t+\Delta t}, \mathbf{V}_{t+\Delta t}]^\top$
10: **end for**
11: **return** $\mathbf{X}_1$

where $\mathcal{R}$ denotes a convolutional residual operator, and AdaGN applies feature-wise affine transformations conditioned on $\mathbf{c}$.

To model long-range temporal dependencies, we incorporate multi-head self-attention (MHSA) (Vaswani et al., 2017) into the latent dynamics. The attention-enhanced representation is combined with a residual connection, normalized, and linearly projected to parameterize the acceleration field:

$$\mathbf{U}_t = \text{Conv1D}_\psi\Big(\text{LN}\big(\mathbf{H}^{(N)} + \text{MHSA}(\mathbf{H}^{(N)})\big)\Big), \quad (10)$$

where $\mathbf{U}_t \in \mathbb{R}^{T_F \times D}$ defines the acceleration in the second-order ODE.

### 4.4. Second-Order Flow Matching and Solver

We define the target phase-space state as $\mathbf{S}_1 = [\mathbf{X}_1, \mathbf{V}_1]^\top$, where $\mathbf{X}_1 = \boldsymbol{x}^F$ denotes the ground-truth future positions. Given the initial state $\mathbf{S}_0 = [\mathbf{X}_0, \mathbf{V}_0]^\top$, the target velocity is derived under a constant-acceleration assumption over $t \in [0, 1]$:

$$\mathbf{V}_1 = \mathbf{V}_0 + 2(\mathbf{X}_1 - \mathbf{X}_0 - \mathbf{V}_0). \qquad (11)$$

Following Conditional Flow Matching, training is conducted along a linear Optimal Transport (OT) path in the augmented phase space,

$$\mathbf{S}_t = (1 - t)\mathbf{S}_0 + t\mathbf{S}_1, \qquad (12)$$

which corresponds to a constant acceleration in the underlying kinematic system. Accordingly, the Neural Acceleration Field $\mathcal{A}_\theta$ is trained to regress the implied acceleration:

$$\mathcal{L}(\theta) = \mathbb{E}_{t, \mathbf{S}_0, \mathbf{S}_1}\big[\|\mathcal{A}_\theta(\mathbf{S}_t, t, \mathbf{e}) - (\mathbf{V}_1 - \mathbf{V}_0)\|^2\big]. \quad (13)$$

At inference, the learned acceleration field is deployed within a second-order dynamical system. We discretize $t \in [0, 1]$ into $K$ steps and evolve the system using a sym-

plectic Euler integrator (Zhong et al., 2020):

$$\mathbf{V}_{t+\Delta t} = \mathbf{V}_t + \mathcal{A}_\theta(\mathbf{S}_t, t, \mathbf{e})\,\Delta t, \qquad (14)$$

$$\mathbf{X}_{t+\Delta t} = \mathbf{X}_t + \mathbf{V}_{t+\Delta t}\,\Delta t, \qquad (15)$$

where $\Delta t = 1/K$. Despite being trained on linear OT paths, KineFlow generates smooth and physically consistent trajectories at inference. The full procedure is summarized in Algorithm 1.

## 5. Theoretical Analysis

We provide a theoretical analysis demonstrating that Kine-Flow preserves flow-matching consistency, attenuates high-frequency errors via double integration, and resolves phase-space ambiguities inherent to first-order dynamics.

### 5.1. Consistency of Phase-Space Flow Matching

Unlike standard flow matching, our formulation introduces a kinematic inductive bias via a structured phase-space parameterization, modeling dynamics as a force-driven second-order system.

Despite this structure, our approach preserves the consistency guarantees of standard flow matching: if the acceleration field $\mathcal{A}_\theta$ minimizes the flow-matching objective in the augmented phase space, the induced vector field

$$\mathbf{u}(\mathbf{S}, t) = \begin{bmatrix} \mathbf{V} \\ \mathcal{A}_\theta(\mathbf{S}, t) \end{bmatrix}$$

satisfies the continuity equation on $\mathbb{R}^{2D}$, ensuring that the marginal position distribution at $t = 1$ matches the target (Lipman et al., 2023). The proof is provided in Appendix A.1.

### 5.2. Kinematic Smoothing

Despite the implicit spectral bias of neural networks (Rahaman et al., 2019), generative models may still exhibit high-frequency artifacts. Second-order integration introduces an explicit kinematic smoothing effect, effectively attenuating high-frequency components in the generated trajectories.

**Assumption 5.1.** The learned vector field contains an additive, zero-mean error $\epsilon(t)$ with predominantly high-frequency content ($\omega \gg 1$), and bounded magnitude $|\epsilon(t)| \le \epsilon_{\max}$.

**Proposition 5.2.** Let $E^{(1)}(t)$ and $E^{(2)}(t)$ denote the position errors induced by $\epsilon(t)$ in first-order (velocity-based) and second-order (acceleration-based) flows. Their spectral magnitudes satisfy

$$|\hat{E}^{(1)}(\omega)| \propto \omega^{-1}, \quad |\hat{E}^{(2)}(\omega)| \propto \omega^{-2}.$$

*Proof Sketch.* Integration in time attenuates high-frequency components: first-order flows reduce them by $\omega^{-1}$, while second-order flows further attenuate them by $\omega^{-2}$, yielding stronger smoothing. See Appendix A.2 for details. □

### 5.3. Resolution of Phase-Space Ambiguities

In time-series forecasting, intersecting trajectories, where identical current states can evolve into different future trends (e.g., regime crossing or trend reversal), cannot be represented by a first-order vector field without singularities.

**Proposition 5.3.** *At a spacetime point $(\mathbf{x}^*, t^*)$ with two velocities $\mathbf{v}_1 \ne \mathbf{v}_2$, any first-order vector field $\dot{\mathbf{x}} = f(\mathbf{x}, t)$ is ill-defined.*

*Proof.* A well-defined ODE requires a unique vector at each point. Having $f(\mathbf{x}^*, t^*) = \mathbf{v}_1$ and $f(\mathbf{x}^*, t^*) = \mathbf{v}_2$ is a contradiction. Lifting to phase space $\mathbf{S} = [\mathbf{x}, \mathbf{v}]$ resolves this: $\mathbf{S}_1 \ne \mathbf{S}_2$, allowing a valid acceleration mapping $\mathcal{A}(\mathbf{S}, t)$. □

### 5.4. Robustness to Field Perturbations

We analyze the resilience of the generation process to transient irregularities in the learned vector field, such as those induced by noisy conditioning.

**Proposition 5.4** (Inertial Buffering). *Suppose the vector field is perturbed by a brief impulse $\delta(t)$ of magnitude $\Delta$ over a duration $\tau \ll 1$, under the assumption that the flow is approximately linear within $[t, t + \tau]$. The resulting displacement error scales as $O(\Delta\tau)$ for first-order models and as $O(\Delta\tau^2)$ for the second-order our method.*

*Proof Sketch.* In first-order dynamics, the displacement error accumulates linearly, Error $\approx \int \delta\, dt \sim O(\Delta\tau)$. In our method, the perturbation acts at the acceleration level, producing a velocity error of $O(\Delta\tau)$ and a position error of $O(\Delta\tau^2)$ after double integration. For short-lived disturbances, this quadratic dependence substantially attenuates the effect. A full derivation is provided in Appendix A.3. □

## 6. Experiments

### 6.1. Datasets and Baselines

**Datasets.** We evaluate kinematic generalization across diverse benchmarks: (a) **Standard Benchmarks** (Lai et al., 2018; Zhou et al., 2021; Wu et al., 2021): **Traffic**, **Electricity**, **Weather**, **ETT**, and **Exchange**. These datasets reflect relatively stable dynamics with periodicity. (b) **High-Volatility Industrial Datasets** (Wang et al., 2024a): the **AEMO** electricity demand datasets, collected by the Australian Energy Market Operator and covering five regions (TAS, VIC, NSW, QLD, SA). Characterized by sharp peaks

*Table 1.* Forecasting results of models on representative standard benchmarks. History length $T_H$ is 96 and corresponding prediction lengths $T_F$ include $\{96, 192, 336, 720\}$. A lower MSE, CRPS or V-Err indicates a better performance. Averaged results of four prediction lengths are reported here. Full results on all datasets and prediction lengths are provided in Appendix Table 12 and 13.

| | Electricity | | | Weather | | | Traffic | | | ETTm1 | | | AVG | | |
|---|---|---|---|---|---|---|---|---|---|---|---|---|---|---|---|
| | MSE↓ | CRPS↓ | V-Err↓ | MSE↓ | CRPS↓ | V-Err↓ | MSE↓ | CRPS↓ | V-Err↓ | MSE↓ | CRPS↓ | V-Err↓ | MSE↓ | CRPS↓ | V-Err↓ |
| **AF.** (2021) | 0.214 | 0.287 | 0.065 | 0.333 | 0.365 | 0.080 | 0.617 | 0.633 | 0.371 | 0.514 | 0.293 | 0.386 | 0.420 | 0.395 | 0.225 |
| **PTST.** (2023) | 0.249 | 0.280 | 0.071 | 0.310 | 0.305 | 0.072 | 0.448 | 0.583 | 0.375 | 0.444 | 0.274 | 0.374 | 0.363 | 0.361 | 0.223 |
| **DLinear** (2023) | 0.259 | 0.283 | 0.070 | 0.326 | 0.340 | 0.075 | 0.516 | 0.584 | 0.373 | 0.456 | 0.284 | 0.379 | 0.389 | 0.373 | 0.224 |
| **iTrans.** (2024) | 0.178 | 0.285 | 0.074 | 0.260 | 0.346 | 0.078 | 0.427 | 0.585 | 0.376 | 0.409 | 0.272 | 0.381 | 0.319 | 0.372 | 0.227 |
| **TimeXer** (2024b) | 0.171 | 0.268 | 0.067 | 0.240 | 0.330 | 0.083 | 0.467 | 0.594 | 0.381 | **0.382** | 0.262 | 0.367 | 0.315 | 0.363 | 0.225 |
| **CSDI** (2021) | 0.343 | 0.268 | 0.069 | 0.472 | 0.323 | 0.080 | 0.987 | 0.582 | 0.394 | 0.822 | 0.263 | 0.371 | 0.656 | 0.359 | 0.229 |
| **TimeGrad** (2021) | 0.334 | 0.267 | 0.068 | 0.470 | 0.315 | 0.076 | 1.003 | 0.568 | 0.389 | 0.812 | 0.256 | 0.368 | 0.655 | 0.352 | 0.225 |
| **TimeDiff** (2023) | 0.333 | 0.269 | 0.070 | 0.471 | 0.318 | 0.081 | 1.006 | 0.569 | 0.391 | 0.806 | 0.254 | 0.366 | 0.654 | 0.353 | 0.227 |
| **DTS.** (2024) | 0.330 | 0.267 | 0.069 | 0.428 | 0.311 | 0.075 | 0.973 | 0.530 | 0.373 | 0.693 | 0.242 | 0.356 | 0.606 | 0.338 | 0.218 |
| **TSFlow** (2025) | 0.329 | 0.269 | 0.070 | 0.430 | 0.289 | 0.076 | 0.975 | 0.527 | 0.375 | 0.695 | 0.243 | 0.354 | 0.607 | 0.332 | 0.219 |
| **KineFlow** (Ours) | **0.160** | **0.256** | **0.060** | **0.186** | **0.181** | **0.064** | **0.332** | **0.514** | **0.343** | 0.397 | **0.221** | **0.323** | **0.269** | **0.293** | **0.197** |
| **Rel. Improv.** | 6.4% | 2.3% | 3.2% | 22.5% | 37.4% | 11.1% | 22.2% | 2.5% | 8.0% | -3.9% | 8.7% | 8.8% | 14.7% | 11.7% | 9.6% |

*Table 2.* Performance comparison on the AEMO dataset. History length $T_H$ is 24 and prediction lengths $T_F$ include $\{12, 24, 48\}$. We report the results averaged over all five regions and all prediction lengths. Full results are provided in Appendix Table 14.

| Metric | Discriminative Methods | | | | | Generative Methods | | | | | Our Method |
|---|---|---|---|---|---|---|---|---|---|---|---|
| | AF. | PTST. | DLinear | iTrans. | TimeXer | CSDI | T-Grad | T-Diff | DTS | TSFlow | **KineFlow** |
| | (2021) | (2023) | (2023) | (2024) | (2024b) | (2021) | (2021) | (2023) | (2024) | (2025) | (Ours) |
| MSE(↓) | 0.957 | 0.963 | 0.822 | 0.746 | 0.628 | 1.067 | 0.999 | 1.021 | 0.850 | 0.818 | **0.400** (36.3% ↓) |
| CRPS(↓) | 0.662 | 0.600 | 0.612 | 0.598 | 0.629 | 0.532 | 0.533 | 0.528 | 0.511 | 0.498 | **0.459** (7.8% ↓) |
| V-Err(↓) | 0.236 | 0.240 | 0.249 | 0.223 | 0.219 | 0.201 | 0.204 | 0.200 | 0.190 | 0.187 | **0.146** (21.9% ↓) |

and rapid momentum shifts, the AEMO datasets stringently test the model's ability to capture high-frequency dynamics.

**Baselines.** We compare KineFlow against a comprehensive suite of state-of-the-art models: (1) **Discriminative methods**: Autoformer (Wu et al., 2021), PatchTST (Nie et al., 2023), DLinear (Zeng et al., 2023), iTransformer (Liu et al., 2024), and TimeXer (Wang et al., 2024b). (2) **Generative methods**: CSDI (Tashiro et al., 2021), TimeGrad (Rasul et al., 2021), TimeDiff (Kollovieh et al., 2023), Diffusion-TS (Yuan & Qiao, 2024), and TSFlow (Kollovieh et al., 2025). Method details are in the related work section.

### 6.2. Evaluation Metrics

**Point forecasting (MSE ↓).** We measure deterministic accuracy using Mean Squared Error (MSE). Point predictions are generated via deterministic inference from a fixed initial state ($\sigma = 0$ in Eq. 5).

**Probabilistic Quality (CRPS ↓).** We assess distributional coverage using the Continuous Ranked Probability Score (CRPS) (Gneiting & Raftery, 2007). We sample 100 trajectories by drawing the initial state $\mathbf{S}_0 \sim \mathcal{N}(\boldsymbol{\mu}_0, \sigma^2\mathbf{I})$ in Eq. 5 and propagating them through the generative model to measure the discrepancy against ground truth.

**Kinematic Consistency (V-Err↓).** We introduce Velocity Error (V-Err) to quantify errors in first-order temporal gradients:

$$\text{V-Err} = \frac{1}{P} \sum_{i,t} \left| (\hat{y}_t - \hat{y}_{t-1}) - (y_t - y_{t-1}) \right|, \qquad (16)$$

where $P$ is the total number of predictions. Lower V-Err indicates better preservation of kinematic momentum and reduced high-frequency noise. Details of all metrics are provided in Appendix D.

### 6.3. Implementation Details

To ensure a fair comparison, all methods are given the identical raw observational data. While baselines typically ingest concatenated raw inputs, KineFlow aligns them into endogenous histories $\boldsymbol{x}^H$ and exogenous forcings $\boldsymbol{e}$. Exact feature partitions are provided in Appendix B. All experiments were conducted on a single NVIDIA H100 GPU. We adopt a two-stage training schedule: the backbone $\mathcal{F}_\phi$ is trained for 30 epochs and then frozen, after which the acceleration field $\mathcal{A}_\theta$ is optimized for 15 epochs using Adam ($\eta = 10^{-4}$, $B = 32$). The model is lightweight ($d = 128$ in Eq. 8, $N = 2$ residual blocks, and $h = 4$ MHSA heads in Eq. 10), with 0.90M total parameters. Although phase-space lifting doubles the state dimension ($D \to 2D$), runtime is domi-

nated by temporal attention, which scales as $O(T_F^2)$ with the prediction horizon. For inference, we use a second-order solver with $\Delta t = 0.05$ and $K = 20$ function evaluations, while Diffusion-TS requires $K = 500$ denoising steps. Full efficiency comparisons are provided in Appendix C.

### 6.4. Main Results on Time Series Benchmarks

**Results on standard benchmarks**. As shown in Table 1, KineFlow achieves state-of-the-art performance on the majority of benchmarks. Compared to the best-performing discriminative methods TimeXe, KineFlow achieves a 15% reduction in average MSE (0.315 vs. 0.269), while simultaneously surpassing the generative models TSFlow and DTS. by 12% in CRPS (0.293 vs. 0.332) and 10% in V-Err (0.197 vs. 0.218). The improvements are particularly pronounced on high-volatility datasets (e.g., Weather, Traffic), with MSE reductions up to 23%, demonstrating that KineFlow captures rapid temporal shifts and extreme events that other baselines fail to resolve.

We further evaluate long-term forecasting stability by analyzing horizon-dependent error growth in Figure 3. As the prediction horizon extends from 96 to 720, baselines increases MSE by 68-82%, whereas KineFlow's MSE rises by 65%, demonstrating that the learned second-order inertia suppresses trajectory divergence in long-term forecasting.

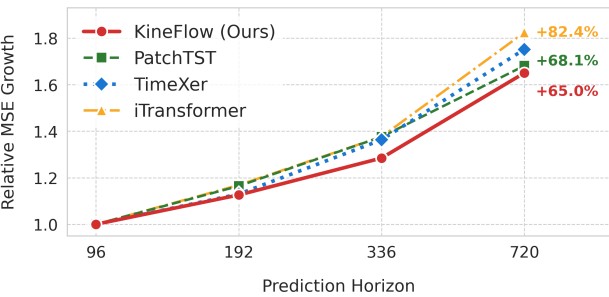

*Figure 3.* Relative MSE growth averaged across all benchmark datasets. Values are normalized with respect to each model's own performance at $T_F = 96$ (set to 1.0).

**Results on High-Volatility Industrial Datasets**. Table 2 reports the results on the AEMO electricity datasets. On average, our method achieves a 36% reduction in MSE (0.400 vs. 0.628) and an 8% in CRPS (0.459 vs. 0.498) compared to the second-best models. Meanwhile, KineFlow outperforms all baselines in V-Err with an average reduction of 22% (0.146 vs. 0.187), which indicates that by integrating the Neural Acceleration Field, our method accurately models the inertia and momentum of electricity demand.

### 6.5. Comparison to Foundation Models

We evaluate the generalizability of KineFlow on the cross-domain transfer forecasting task, where the model is trained

*Table 3.* Cross-domain transfer forecasting performance on AEMO. Results are averaged over prediction lengths $\{12, 24, 48\}$. Foundation models utilize a context length of $T_H = 1000$ (Woo et al., 2024), whereas KineFlow use history length of $T_H = 24$.

| Model | VIC → SA | | | TAS → NSW | | |
|---|---|---|---|---|---|---|
| | MSE | CRPS | V-Err | MSE | CRPS | V-Err |
| **TimesFM** (2024) | 0.372 | 0.430 | 0.255 | 1.115 | 0.599 | 0.221 |
| **Chronos** (2024) | 0.362 | 0.439 | 0.233 | 0.986 | 0.604 | 0.227 |
| **Moirai** (2024) | 0.340 | 0.435 | 0.221 | 0.912 | 0.590 | 0.216 |
| **Timer-XL**(2025a) | 0.347 | 0.433 | 0.225 | 0.885 | 0.603 | 0.220 |
| **Time-MoE**(2025) | 0.315 | 0.433 | 0.205 | 0.910 | 0.597 | 0.218 |
| **Sundial** (2025b) | 0.285 | 0.419 | 0.206 | 0.805 | 0.595 | 0.221 |
| **KineFlow** (Ours) | **0.246** | **0.415** | **0.199** | **0.741** | **0.554** | **0.181** |
| **Rel. Improv.** | 13.7% | 1.0% | 2.9% | 7.9% | 6.1% | 16.2% |

on a source domain (e.g., VIC) and adapted to a target domain (e.g., SA). We compare against large foundation models pre-trained on billions of tokens. Specifically, foundation models are evaluated in a zero-shot setting with a long historical context ($T_H = 1000$), while KineFlow performs few-shot adaptation using only 3% of target-domain data and a much shorter history ($T_H = 24$).

As shown in Table 3, KineFlow consistently outperforms massive foundation models, achieving an 8–14% reduction in MSE relative to Sundial (Liu et al., 2025b), despite using $140\times$ fewer parameters (0.90M vs. 128M). These results suggest that explicitly modeling invariant second-order kinematics provides a more effective inductive bias for cross-domain transfer than scale-driven pattern memorization, enabling strong generalization with minimal data and computational cost.

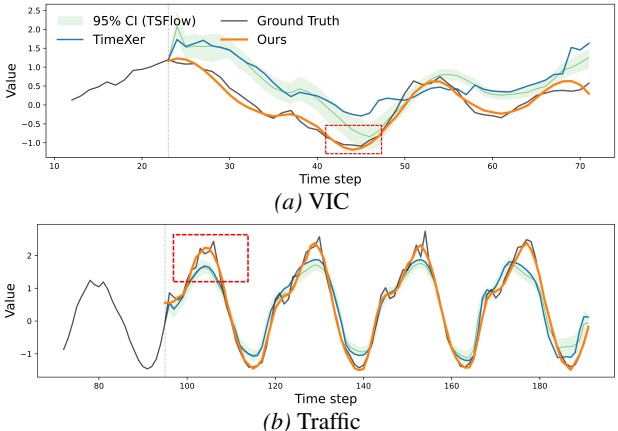

*Figure 4.* Visualizations of forecasting result on different datasets.

### 6.6. Visual Analysis and Case Studies

**Time-Domain Visualization.** Figure 4 compares the forecasts on VIC and Traffic, focusing on high-volatility, non-stationary intervals. Competitive baselines, including

*Table 4.* Ablation studies with respect to architectural enhancements. Gaussian Init: initializing $\mathbf{S}_0$ with noise. 1st-Order: removing the velocity $\mathbf{V}$. Zero-$\mathbf{V}_0$: initializing velocity $\mathbf{V}_0$ with zero.

| Method Variant | ETTm1 | | | Weather | | | VIC | | | SA | | | AVG | | |
|---|---|---|---|---|---|---|---|---|---|---|---|---|---|---|---|
| | MSE | CRPS | V-Err | MSE | CRPS | V-Err | MSE | CRPS | V-Err | MSE | CRPS | V-Err | MSE | CRPS | V-Err |
| Noise Init *(w/o Prior)* | 0.417 | 0.250 | 0.388 | 0.217 | 0.222 | 0.157 | 0.590 | 0.531 | 0.185 | 0.171 | 0.359 | 0.197 | 0.346 | 0.341 | 0.232 |
| 1st-Order *(w/o Velocity)* | 0.422 | 0.230 | 0.367 | 0.195 | 0.205 | 0.102 | 0.532 | 0.535 | 0.199 | 0.182 | 0.377 | 0.186 | 0.333 | 0.337 | 0.214 |
| Zero-$\mathbf{V}_0$ *(w/o Momentum)* | 0.400 | 0.225 | 0.331 | 0.211 | 0.188 | 0.067 | 0.495 | 0.530 | 0.154 | 0.137 | 0.377 | 0.186 | 0.311 | 0.330 | 0.185 |
| KineFlow (Full Model) | **0.397** | **0.221** | **0.323** | **0.186** | **0.181** | **0.064** | **0.490** | **0.529** | **0.154** | **0.134** | **0.350** | **0.121** | **0.302** | **0.320** | **0.166** |

TimeXer (blue) and TSFlow (green), exhibit noticeable phase lag and attenuated responses near sharp extrema, resulting in overly smoothed predictions that underestimate both peaks and troughs. In contrast, KineFlow (orange) more fits the ground truth, preserving both the timing and magnitude of extreme events, which indicate that modeling second-order dynamics leads to more stable temporal evolution under abrupt regime changes.

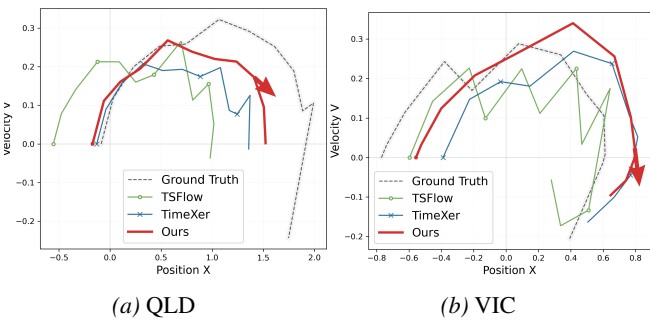

*(a)* QLD       *(b)* VIC

*Figure 5.* Forecasting trajectories in phase space ($\mathbf{X} - \mathbf{V}$).

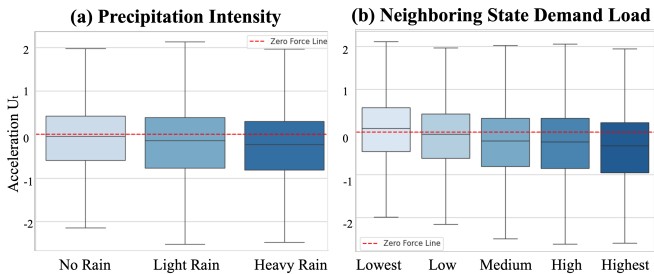

*Figure 6.* Visualization of inferred acceleration distributions conditioned on precipitation intensity and neighboring demand.

**Phase-Space Dynamics Visualization**. Figure 5 shows forecasting trajectories in phase space ($\mathbf{X}$–$\mathbf{V}$). The ground truth (dashed lines) exhibits smooth, coherent trajectories, whereas TimeXer (blue) and TSFlow (green) generate irregular paths with abrupt directional changes, indicating a failure to capture the continuous temporal evolution of the underlying system. In contrast, KineFlow (red) yields smooth trajectories that closely align with the ground truth, indicating that the second-order formulation enforces temporal continuity and stabilizes momentum evolution.

**Physical interpretability analysis**. To assess whether the learned dynamics align with meaningful physical mechanisms rather than spurious correlations, we examine the inferred acceleration $\mathbf{a}$ conditioned on exogenous drivers in Fig. 6. As shown in Fig. 6(a), increasing precipitation intensity leads to a clear negative shift in acceleration, which indicates that KineFlow correctly identifies rainfall as a factor that dampens the electricity demand. Similarly, Fig. 6(b) shows that the acceleration systematically varies with neighboring state demand, indicating that our method effectively captures the influence of inter-regional coupling.

## 6.7. Ablation Studies

We conduct ablation studies to isolate the contributions of KineFlow's core components in Table 4.

**Effectiveness of Phase-Space Prior Construction**. We replace the informative initialization $\mathbf{S}_0 = [\mathbf{X}_0, \mathbf{V}_0]^\top$ with Gaussian noise $\mathcal{N}(\mathbf{0}, \mathbf{I})$, while retaining $\mathcal{F}_\phi(\boldsymbol{x}^H)$ and historical momentum as conditional inputs to $\mathcal{A}_\theta$. As shown in Table 4, the informative prior yields a 13% reduction in MSE (0.302 vs. 0.346) and a 28% drop in V-Err (0.166 vs. 0.232), indicating that initializing meaningful state enables the model to focus on refining temporal dynamics rather than reconstructing trajectories from scratch.

**Impact of Second-Order Dynamics**. We benchmarked against a first-order baseline in which the state $\mathbf{S}$ contains position $\mathbf{X}$ only , removing the velocity $\mathbf{V}$. As detailed in Table 4, KineFlow reduces average MSE by 9% (0.302 vs. 0.333) , CRPS by 5% (0.320 vs. 0.337) and V-Err by 22% (0.166 vs. 0.214). The pronounced improvement highlights the importance of explicitly modeling acceleration and inertia, enabling smoother temporal evolution and improved physical consistency compared to first-order dynamics.

**Impact of Initial Velocity**. We further evaluate a Zero-$\mathbf{V}_0$ baseline that initializes velocity $\mathbf{V}_0$ to zero, while KineFlow is initialized as Eq. 7. Table 4 shows that KineFlow achieves average reductions of 3% (0.302 vs. 0.311) in MSE, 3% in CRPS (0.320 vs. 0.330), and 10% in V-Err (0.166 vs. 0.185), demonstrating that initializing $\mathbf{V}_0$ from historical momentum yields a more physically consistent starting state and improves alignment with past trends.

# 7. Conclusion

We propose KineFlow, a generative forecasting framework that models time-series dynamics in second-order phase space to address kinematic inconsistencies in long-horizon prediction. By learning a neural acceleration field via flow matching, KineFlow enforces physical continuity and resolves phase-space ambiguities in first-order models. Experiments on standard benchmarks and high-volatility industrial datasets demonstrate state-of-the-art accuracy, improved stability, and interpretable dynamics, while achieving strong few-shot cross-domain transfer with orders of magnitude fewer parameters than foundation models. We hope this work can inspire future generative forecasting paradigms that integrate structured temporal dynamics into probabilistic modeling.

# Acknowledgements

This work was supported in part by the National Key R&D Program of China (Grant No.2023YFF0725001), in part by the National Natural Science Foundation of China(Grant No.92370204),in part by the guangdong Basic and Applied Basic Research Foundation (Grant No.2023B1515120057), in part by the Key-Area Special Project of Guangdong Provincial Ordinary Universities (2024ZDZX1007)

# Impact Statement

Time-series forecasting is a cornerstone of decision-making in critical sectors, including energy, climate science, and public infrastructure. This work bridges deep generative modeling and physical principles by demonstrating that kinematic inductive biases can overcome the limitations of purely data-driven approaches, inspiring further research in physically consistent generative AI. Our approach is accurate, interpretable, and environmentally efficient, aligning with Green AI principles. We provide code to support reproducibility and emphasize that this work focuses on scientific advancement without foreseeable negative social impacts.

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

# A. Detailed Theoretical Proofs

### A.1. Proof of Proposition 5.1

We provide the detailed derivation showing that minimizing the acceleration matching objective corresponds to minimizing the standard CFM objective under the proposed structured parameterization in the augmented phase space.

**Setup.** We assume a differentiable probability path in the augmented phase space, with well-defined velocity and acceleration components, consistent with the conditional paths used in CFM. Let $\mathbf{S} = [\mathbf{X}, \mathbf{V}]^\top \in \mathbb{R}^{2D}$ denote the augmented phase-space state. Standard CFM aims to learn a time-dependent vector field $\mathbf{v}_\theta$ that approximates the conditional vector field $\mathbf{u}_t$ generating the probability path $p_t(\mathbf{S})$. The training objective is to minimize the regression loss:

$$\mathcal{L}(\theta) = \mathbb{E}_{t,q(\mathbf{S}_0),\mathbf{S}_1} \left[ \|\mathbf{v}_\theta(\mathbf{S}_t, t) - \mathbf{u}_t(\mathbf{S}_t|\mathbf{S}_1)\|^2 \right], \tag{17}$$

where $\mathbf{v}_\theta(\cdot, t)$ is the neural vector field parameterized by $\theta$, and $\mathbf{u}_t(\cdot|\mathbf{S}_1)$ represents the target vector field conditioned on the endpoint $\mathbf{S}_1$.

**Phase-Space Decomposition.** In the KineFlow framework, the vector fields acting on the augmented state $\mathbf{S} = [\mathbf{X}, \mathbf{V}]^\top$ possess a rigid block structure imposed by kinematic laws.

1. **Target Vector Field ($\mathbf{u}_t$):** We define the target vector field component-wise. For the velocity component, we assign the current state velocity $\mathbf{V}_t$ to enforce kinematic consistency. For the acceleration component, we utilize the time-derivative of the velocity path:

$$\mathbf{u}_t(\mathbf{S}_t|\mathbf{S}_1) := \begin{bmatrix} \mathbf{V}_t \\ \frac{d}{dt}\mathbf{V}_t \end{bmatrix} = \begin{bmatrix} \mathbf{V}_t \\ \mathbf{a}_t \end{bmatrix}.$$

2. **Neural Vector Field ($\mathbf{v}_\theta$):** We parameterize the learned vector field to strictly respect the phase-space topology:

$$\mathbf{v}_\theta(\mathbf{S}_t, t) = \begin{bmatrix} \mathbf{V}_t \\ \mathcal{A}_\theta(\mathbf{S}_t, t) \end{bmatrix}.$$

Here, the position update (first component) is *structurally constrained* to be $\mathbf{V}_t$, containing no learnable parameters. The network $\mathcal{A}_\theta$ only learns the acceleration component.

**Loss Simplification.** Substituting these structured forms into the generic Flow Matching objective yields:

$$\mathcal{L}_{\text{FM}}(\theta) = \mathbb{E}\left[ \|\mathbf{v}_\theta(\mathbf{S}_t, t) - \mathbf{u}_t(\mathbf{S}_t|\mathbf{S}_1)\|^2 \right] \tag{18}$$

$$= \mathbb{E}\left[ \left\| \begin{bmatrix} \mathbf{V}_t \\ \mathcal{A}_\theta(\mathbf{S}_t, t) \end{bmatrix} - \begin{bmatrix} \mathbf{V}_t \\ \mathbf{a}_t \end{bmatrix} \right\|^2 \right] \tag{19}$$

$$= \mathbb{E}\left[ \underbrace{\|\mathbf{V}_t - \mathbf{V}_t\|^2}_{\text{Kinematic Constraint} \equiv 0} + \|\mathcal{A}_\theta(\mathbf{S}_t, t) - \mathbf{a}_t\|^2 \right] \tag{20}$$

$$= \mathbb{E}\left[ \|\mathcal{A}_\theta(\mathbf{S}_t, t) - \mathbf{a}_t\|^2 \right] = \mathcal{L}_{\text{acc}}(\theta). \tag{21}$$

**Conclusion.** Under the proposed structured phase-space parameterization, the generic flow-matching objective reduces exactly to the acceleration matching loss $\mathcal{L}_{\text{acc}}(\theta)$. Since the position dynamics $\dot{\mathbf{X}} = \mathbf{V}$ are satisfied by construction, optimizing $\mathcal{L}_{\text{acc}}(\theta)$ yields a vector field that satisfies the continuity equation in the augmented space. By the standard consistency guarantees of CFM (Lipman et al., 2023), the induced dynamics recover the target marginal distribution at $t = 1$. $\qquad\square$

### A.2. Proof of Proposition 5.2

We provide a rigorous derivation of the spectral decay properties based on linear perturbation analysis.

**Setup.** Let $\mathbf{x}^*(t)$ denote the noiseless trajectory and $\mathbf{x}(t)$ the trajectory under additive noise $\epsilon(t)$. The error is defined as $E(t) = \mathbf{x}(t) - \mathbf{x}^*(t)$. We assume zero initial conditions $E(0) = \dot{E}(0) = 0$ to isolate the effect of $\epsilon(t)$.

**First-Order Baseline.** Consider a first-order flow where the velocity estimate is perturbed by noise: $\dot{\mathbf{x}}(t) = v(t) + \epsilon(t)$. Subtracting the noiseless baseline $\dot{\mathbf{x}}^*(t) = v(t)$ yields the error dynamics:

$$\dot{E}^{(1)}(t) = \epsilon(t). \tag{22}$$

Taking the Fourier transform $\mathcal{F}\{\cdot\}$ of both sides, and utilizing the differentiation property $\mathcal{F}\{\dot{f}\} = i\omega \hat{f}(\omega)$:

$$i\omega \hat{E}^{(1)}(\omega) = \hat{\epsilon}(\omega) \tag{23}$$

$$\hat{E}^{(1)}(\omega) = (i\omega)^{-1}\hat{\epsilon}(\omega). \tag{24}$$

The spectral magnitude decays as:

$$|\hat{E}^{(1)}(\omega)| = \omega^{-1}|\hat{\epsilon}(\omega)|. \tag{25}$$

**Second-Order KineFlow (Ours).** For our method , the acceleration estimate is perturbed: $\ddot{\mathbf{x}}(t) = \mathcal{A}(t) + \epsilon(t)$. Subtracting the noiseless baseline $\ddot{\mathbf{x}}^*(t) = \mathcal{A}(t)$ yields the second-order error dynamics:

$$\ddot{E}^{(2)}(t) = \epsilon(t). \tag{26}$$

Applying the Fourier transform (where the second derivative corresponds to $(i\omega)^2$):

$$(i\omega)^2 \hat{E}^{(2)}(\omega) = \hat{\epsilon}(\omega) \tag{27}$$

$$-\omega^2 \hat{E}^{(2)}(\omega) = \hat{\epsilon}(\omega) \tag{28}$$

$$\hat{E}^{(2)}(\omega) = -\omega^{-2}\hat{\epsilon}(\omega). \tag{29}$$

The spectral magnitude decays as:

$$|\hat{E}^{(2)}(\omega)| = \omega^{-2}|\hat{\epsilon}(\omega)|. \tag{30}$$

**Comparison.** Comparing the asymptotic behavior for high-frequency components ($\omega \gg 1$):

$$\frac{|\hat{E}^{(2)}(\omega)|}{|\hat{E}^{(1)}(\omega)|} = \frac{\omega^{-2}}{\omega^{-1}} = \omega^{-1}. \tag{31}$$

This confirms that our method acts as a stronger low-pass filter, suppressing high-frequency perturbations more aggressively, with an error decay rate of $O(\omega^{-2})$ compared to $O(\omega^{-1})$ in first-order models. $\square$

### A.3. Proof of Proposition 5.4

Let the perturbation $\delta(t)$ be active on $t \in [t_0, t_0 + \tau]$. For our method , the position error is derived by double integration:

$$
\begin{aligned}
E^{(2)}(t_0 + \tau) &= \int_{t_0}^{t_0+\tau} \left( \int_{t_0}^{s} \delta(u)\, du \right) ds \\
&\leq \int_{t_0}^{t_0+\tau} (\Delta \cdot (s - t_0))\, ds \\
&= \Delta \left[ \frac{1}{2}(s - t_0)^2 \right]_{t_0}^{t_0+\tau} = \frac{1}{2}\Delta\tau^2.
\end{aligned}
$$

In contrast, for a first-order model, $E^{(1)}(t_0 + \tau) = \int_{t_0}^{t_0+\tau} \delta(u)du = \Delta\tau$. Since $\tau \ll 1$, the term $\tau^2$ is an order of magnitude smaller than $\tau$, demonstrating superior robustness. $\square$

## B. Detailed Feature Configuration

We provide exact specifications of the feature partitioning used in experiments. For each dataset, inputs are split into **Endogenous histories** $\mathbf{x}^H$ and **Exogenous forcing** e. All exogenous features are computed from the original dataset (e.g., first differences, time embeddings) and do not introduce external information.

## B.1. Partitioning Logic per Dataset

Table 5 summarizes the feature partitioning for each dataset. Following our second-order formulation, raw inputs are split into: (i) **endogenous histories** $x^H$, i.e., multivariate past observations used to initialize position and velocity; and (ii) **exogenous forcing** $e$, which conditions the neural acceleration field. The forcing vector consists of deterministic quantities derived from the same raw data and timestamps, including physical drivers (e.g., solar radiation), first differences, and time embeddings.

*Table 5.* Feature partitioning configuration.

| Dataset | Endogenous histories $\mathrm{x}^H$ | Exogenous forcing $\mathrm{e}$ |
|---|---|---|
| **Electricity** (Lai et al., 2018) | Hourly consumption traces for all clients (target included). | Time embeddings (hour-of-day, day-of-week, day-of-year, month). |
| **Weather** (Wu et al., 2021) | Target temperature, dew point, relative humidity and other meteorological variables. | Solar radiation (SWDR), PAR; precipitation; first differences of pressure ($\Delta$Pressure); time embeddings (hour, day, month). |
| **Traffic** (Lai et al., 2018) | Flow rates from all sensors (target included). | Global statistics (mean, std of occupancy); time embeddings (hour-of-day, day-of-week). |
| **ETT** (Zhou et al., 2021) | All monitor channels (oil temperature and 6 load indicators). | First differences of load features (e.g., $\Delta$HUFL) as impulsive forcing; time embeddings (hour, day, month). |
| **Exchange** (Lai et al., 2018) | Daily exchange rates of 8 currencies against USD (target included). | Returns differences (market momentum); calendar embeddings (day-of-week, day-of-year). |
| **Australian Energy (AEMO)** (Wang et al., 2024a) | Half-hourly regional load: target region plus four interconnected regions; calendar flags (workday, holiday). | Temperature deltas ($\Delta$min/max), humidity deltas, pressure deltas; total precipitation; time embeddings. |

## B.2. Input Fairness

As shown in Table 7, simply concatenating exogenous features ($\mathbf{e}$) into TimeXer does not close the performance gap. Even with augmented inputs, TimeXer underperforms KineFlow by a large margin (0.373 vs. 0.302). This result indicates that the benefit of exogenous features arises from their structural integration via the acceleration field, rather than from treating them as additional input channels.

*Table 6.* **Input Fairness Ablation.** Comparison between KineFlow and the strongest baseline (TimeXer) trained with augmented inputs. *Standard*: raw history only. *+Augmented*: raw history concatenated with all exogenous forcing features ($\mathbf{e}$) used in KineFlow.

| Model Configuration | ETTm1 | | | Weather | | | VIC | | | SA | | | AVG | | |
|---|---|---|---|---|---|---|---|---|---|---|---|---|---|---|---|
| | MSE | CRPS | V-Err | MSE | CRPS | V-Err | MSE | CRPS | V-Err | MSE | CRPS | V-Err | MSE | CRPS | V-Err |
| TimeXer *(Standard)* | 0.382 | 0.262 | 0.367 | 0.240 | 0.330 | 0.083 | 0.632 | 0.811 | 0.216 | 0.243 | 0.587 | 0.236 | 0.374 | 0.498 | 0.226 |
| TimeXer *(+Augmented Inputs)* | 0.379 | 0.260 | 0.365 | 0.251 | 0.316 | 0.080 | 0.621 | 0.872 | 0.219 | 0.240 | 0.590 | 0.240 | 0.373 | 0.510 | 0.226 |
| **KineFlow (Ours)** | 0.397 | 0.221 | 0.323 | 0.186 | 0.181 | 0.064 | 0.490 | 0.529 | 0.154 | 0.134 | 0.350 | 0.121 | 0.302 | 0.320 | 0.167 |

We also augmented strong pre-trained baselines (Chronos-2, Timer-XL) with the exact same exogenous features used by KineFlow. As shown in Table 7, simply appending these features yields marginal gains for the foundation models. Under identical input settings, KineFlow still outperforms the strongest augmented baseline (Chronos-2 + Augmented) by 7.2% (VIC $\rightarrow$ SA) and 2.9% (TAS $\rightarrow$ NSW) in MSE. This provides empirical evidence that KineFlow's effectiveness is largely due to its second-order kinematic formulation.

## B.3. Preprocessing Details

To ensure numerical stability in the ODE solver:

- **Endogenous States:** Standardized using Z-score normalization ($\mu = 0, \sigma = 1$) based on the training set statistics.

- **Exogenous Forces:** Physical variables (e.g., radiation) are similarly normalized. Constructed differences (e.g., $\Delta$Temperature) are scaled to match the dynamic range of the state variables.

*Table 7.* Input Fairness Comparison with Foundation Models.

| Model Configuration | VIC → SA | | | TAS →NSW | | |
|---|---|---|---|---|---|---|
| | MSE | CRPS | V-Err | MSE | CRPS | V-Err |
| Chronos-2 *(Standard)* | 0.290 | 0.427 | 0.210 | 0.801 | 0.609 | 0.190 |
| Chronos-2 *(+Augmented Inputs)* | 0.265 | 0.427 | 0.204 | 0.763 | 0.603 | 0.193 |
| Timer-XL *(Standard)* | 0.355 | 0.430 | 0.229 | 0.881 | 0.593 | 0.205 |
| Timer-XL *(+Augmented Inputs)* | 0.347 | 0.433 | 0.225 | 0.885 | 0.603 | 0.220 |
| **KineFlow (Ours)** | **0.246** | **0.415** | **0.199** | **0.741** | **0.554** | **0.181** |

## B.4. Dataset Statistics

Table 8 summarizes the statistical characteristics of all benchmarks, including the dimensionality of the constructed **endogenous states** ($D$) and **exogenous forcing** ($D_e$).

*Table 8.* Details of the dataset statistics.

| Dataset | Frequency | Total Steps | Prediction Task $(T_H \to T_F)$ | Feature Dimensions | |
|---|---|---|---|---|---|
| | | | | Endogenous ($D$) | Exogenous ($D_e$) |
| **Australian Energy** (AEMO) | 30 min | 70,128 | $24 \to 12/24/48$ | 7 | 11 |
| **Traffic** | 1 hour | 17,544 | $96 \to 96/192/336/720$ | 862 | 6 |
| **Electricity** (ECL) | 1 hour | 26,304 | $96 \to 96/192/336/720$ | 321 | 4 |
| **Weather** | 10 min | 52,696 | $96 \to 96/192/336/720$ | 21 | 6 |
| **Exchange** | 1 day | 7,588 | $96 \to 96/192/336/720$ | 8 | 12 |
| **ETTh1** | 1 hour | 17,420 | $96 \to 96/192/336/720$ | 7 | 11 |
| **ETTh2** | 1 hour | 17,420 | $96 \to 96/192/336/720$ | 7 | 11 |
| **ETTm1** | 15 min | 69,680 | $96 \to 96/192/336/720$ | 7 | 12 |
| **ETTm2** | 15 min | 69,680 | $96 \to 96/192/336/720$ | 7 | 12 |

## C. Computational Efficiency Profile

We compare the computational efficiency of KineFlow with representative discriminative and generative baselines in Table 9, reporting both model size and inference-time complexity.

Diffusion-based models such as Diffusion-TS require a large number of denoising steps ($K = 500$), resulting in substantial inference overhead. In contrast, KineFlow performs inference with a second-order ODE solver using only $K = 20$ function evaluations, corresponding to a $25\times$ reduction in the number of inference steps. This places KineFlow in a similar latency regime to flow-matching methods such as TSFlow ($K = 32$), while providing improved kinematic consistency.

In terms of parameterization, KineFlow contains 0.90M parameters in total. However, the model is composed of a frozen backbone (0.67M) and a lightweight, trainable acceleration field (0.23M). During training and adaptation, only the acceleration field is optimized, resulting in fewer trainable parameters than Diffusion-TS (0.29M). This design decouples representational capacity from training cost, enabling efficient optimization without increasing inference-time complexity.

## D. Metric

**Probabilistic Forecasting Quality (CRPS)**. We evaluate probabilistic forecasts using the Continuous Ranked Probability Score (CRPS) (Gneiting & Raftery, 2007), which measures the calibration and sharpness of the predicted distribution. For a predictive cumulative distribution function $F$ and observation $x$, CRPS is defined as

$$\text{CRPS}(F, x) = \int_{-\infty}^{\infty} \left(F(y) - \mathbf{1}\{y \geq x\}\right)^2 \, dy, \tag{32}$$

where $\mathbf{1}\{\cdot\}$ denotes the indicator function. CRPS is equivalent to the integrated Brier score over all potential real-valued thresholds. In our implementation, $F$ is represented empirically by an ensemble of 100 Monte Carlo samples generated

*Table 9.* Model Complexity and Inference Cost. Parameter count (in millions) and number of inference steps required for forecasting.

| Method | Model Class | Parameter Count (M) | Inference Step($K$) | Latency (ms/batch) |
|---|---|---|---|---|
| **Autoformer** (Wu et al., 2021) | Transformer | 0.67 | 1 | 2.9 |
| **TimeXer** (Wang et al., 2024b) | Transformer | 0.45 | 1 | 2.2 |
| **Diffusion-TS** (Yuan & Qiao, 2024) | Diffusion | 0.29 | 500 | 620.0 |
| **TSFlow** (Kollovieh et al., 2025) | Flow Matching | 0.18 | 32 | 36.5 |
| **KineFlow** | **2nd-Order FM** | **0.90**[†] | **20** | 22.8 |

[†] Composed of the frozen backbone (0.67M) and the learnable acceleration field (0.23M).

during inference. CRPS penalizes both mean prediction bias and miscalibrated variance, providing a robust measure of distributional fidelity.

**Physical Consistency (V-Err)**. We measure dynamic realism and smoothness with the Variational Error (V-Err), defined as the mean absolute difference between predicted and true one-step velocities:

$$\text{V-Err} = \frac{1}{P} \sum_{i,t} \left| (\hat{y}_t - \hat{y}_{t-1}) - (y_t - y_{t-1}) \right|, \tag{33}$$

where $\hat{y}$ and $y$ denote the predicted and ground-truth values, respectively. A lower V-Err indicates that the model has successfully captured the system's latent momentum and effectively mitigated the starting point fracture or unnatural jitter common in standard generative models.

## E. Robustness Across Diverse Deterministic Backbones

To further assess whether the performance gains introduced by KineFlow are consistent across different backbone architectures, we evaluate our framework using five diverse deterministic models. These models are employed to construct the prior defined in Eq. 6 of the main manuscript.

*Table 10.* KineFlow Performance Across Diverse Deterministic Backbones.

| Backbone Choice | Weather | | | VIC | | |
|---|---|---|---|---|---|---|
| | MSE ↓ | CRPS ↓ | V-Err ↓ | MSE ↓ | CRPS ↓ | V-Err ↓ |
| PatchTST | 0.189 | 0.331 | 0.062 | 1.215 | 0.648 | 0.219 |
| *+ KineFlow* | **0.132** | **0.229** | **0.061** | **0.715** | **0.580** | **0.205** |
| DLinear | 0.276 | 0.332 | 0.065 | 0.854 | 0.693 | 0.271 |
| *+ KineFlow* | **0.157** | **0.244** | **0.065** | **0.699** | **0.595** | **0.200** |
| iTransformer | 0.175 | 0.336 | 0.070 | 0.712 | 0.701 | 0.251 |
| *+ KineFlow* | **0.110** | **0.208** | **0.065** | **0.613** | **0.593** | **0.205** |
| TimeXer | 0.157 | 0.320 | 0.073 | 0.653 | 0.910 | 0.218 |
| *+ KineFlow* | **0.115** | **0.199** | **0.070** | **0.562** | **0.683** | **0.195** |

As shown in Table 10, KineFlow consistently enhances predictive performance across diverse deterministic backbones and high-volatility datasets by leveraging its second-order dynamics as a structural refinement mechanism to correct suboptimal initial trajectories.

## F. Ablation Study on Architecture Design

The integration of 1D Convolution (Conv1D) and Multi-Head Self-Attention (MHSA) in KineFlow is specifically designed to capture complementary aspects of temporal dynamics. Specifically, Conv1D models local kinematic consistency along the forecast horizon, while MHSA captures long-range dependencies to ensure global coherence. As demonstrated in Table 11, ablating either component leads to a notable drop in performance.

*Table 11.* Architecture Ablation.

| Variant | Weather | | VIC | |
|---|---|---|---|---|
| | MSE ↓ | V-Err ↓ | MSE ↓ | V-Err ↓ |
| **KineFlow (Full)** | **0.186** | **0.064** | **0.490** | **0.154** |
| *w/o* Conv1D | 0.192 | 0.069 | 0.581 | 0.179 |
| *w/o* MHSA | 0.195 | 0.079 | 0.553 | 0.172 |

## G. Detailed Forecasting Results

In the main text, we reported the performance averaged over four prediction lengths to conserve space. In this section, we provide the complete, granular performance breakdown for all individual prediction horizons.

Table 12, 13 detail the results on standard benchmarks (ETT, Traffic, Electricity, Weather, Exchange) with horizons $T_F \in \{96, 192, 336, 720\}$. Notably, KineFlow exhibits strong long-term robustness. While baseline performance typically degrades rapidly at $T_F = 720$, our second-order formulation effectively mitigates error accumulation, preserving trajectory smoothness. This is particularly evident in physical domains (Traffic, Electricity), where KineFlow reduces MSE by substantial margins. On the highly stochastic Exchange dataset, KineFlow maintains superior probabilistic calibration (state-of-the-art CRPS) and competitive point forecasting accuracy across extended timeframes.

Table 14 presents the detailed results on the high-volatility AEMO dataset with prediction horizons $T_F \in \{12, 24, 48\}$. Consistent with the averaged results, KineFlow demonstrates superior performance across all horizons.

*Table 12.* Full results on ETT datasets. Comparison of MSE, CRPS, and V-Err on ETTh1, ETTh2, ETTm1, and ETTm2. **Bold** indicates the best result, and underlined indicates the second best.

| Model | Len | ETTh1 | | | ETTh2 | | | ETTm1 | | | ETTm2 | | |
|---|---|---|---|---|---|---|---|---|---|---|---|---|---|
| | | MSE | CRPS | V-Err | MSE | CRPS | V-Err | MSE | CRPS | V-Err | MSE | CRPS | V-Err |
| **AF.** (2021) | 96 | 0.436 | 0.293 | 0.177 | 0.346 | 0.263 | **0.154** | 0.510 | 0.288 | 0.372 | 0.309 | 0.233 | 0.152 |
| | 192 | 0.456 | 0.295 | 0.177 | 0.379 | 0.270 | 0.165 | 0.514 | 0.293 | 0.390 | 0.317 | 0.246 | 0.160 |
| | 336 | 0.485 | 0.299 | 0.178 | 0.401 | 0.273 | 0.193 | 0.510 | 0.295 | 0.391 | 0.325 | 0.247 | 0.160 |
| | 720 | 0.512 | 0.300 | 0.180 | 0.466 | 0.270 | 0.195 | 0.521 | 0.297 | 0.391 | 0.334 | 0.251 | 0.160 |
| **PTST.** (2023) | 96 | 0.471 | 0.279 | 0.175 | 0.352 | 0.257 | 0.160 | 0.395 | 0.274 | 0.374 | 0.295 | 0.224 | 0.150 |
| | 192 | 0.512 | 0.266 | 0.170 | 0.361 | 0.279 | 0.166 | 0.385 | 0.265 | 0.371 | 0.287 | 0.225 | 0.157 |
| | 336 | 0.522 | 0.279 | 0.178 | 0.391 | 0.285 | 0.169 | 0.470 | 0.277 | 0.375 | 0.306 | 0.228 | 0.164 |
| | 720 | 0.647 | 0.280 | 0.173 | 0.400 | 0.288 | 0.193 | 0.525 | 0.279 | 0.375 | 0.310 | 0.235 | 0.177 |
| **DLinear** (2023) | 96 | 0.477 | 0.276 | 0.173 | 0.337 | 0.240 | 0.167 | 0.399 | 0.277 | 0.375 | 0.288 | 0.221 | **0.144** |
| | 192 | 0.465 | 0.281 | 0.175 | 0.371 | 0.253 | 0.179 | 0.435 | 0.280 | 0.377 | 0.293 | 0.225 | 0.157 |
| | 336 | 0.499 | 0.283 | 0.179 | 0.388 | 0.272 | 0.180 | 0.470 | 0.285 | 0.380 | 0.306 | 0.237 | 0.161 |
| | 720 | 0.572 | 0.285 | 0.183 | 0.397 | 0.276 | 0.189 | 0.520 | 0.293 | 0.382 | 0.325 | 0.237 | 0.160 |
| **iTrans.** (2024) | 96 | 0.385 | 0.263 | 0.170 | 0.335 | 0.237 | 0.162 | 0.334 | 0.260 | 0.369 | 0.263 | 0.227 | 0.159 |
| | 192 | 0.442 | 0.263 | 0.172 | 0.395 | 0.262 | 0.180 | 0.387 | 0.265 | 0.371 | 0.271 | 0.238 | 0.165 |
| | 336 | 0.587 | 0.265 | 0.178 | 0.385 | 0.269 | **0.162** | 0.426 | 0.277 | 0.390 | 0.276 | 0.240 | 0.168 |
| | 720 | 0.603 | 0.265 | 0.179 | 0.394 | 0.289 | 0.199 | 0.490 | 0.285 | 0.394 | 0.284 | 0.255 | 0.173 |
| **TimeXer** (2024b) | 96 | **0.382** | 0.260 | 0.166 | **0.295** | 0.221 | 0.157 | 0.319 | 0.257 | 0.366 | **0.257** | 0.224 | 0.155 |
| | 192 | 0.428 | 0.266 | 0.172 | 0.304 | 0.255 | 0.176 | **0.357** | 0.259 | 0.366 | 0.266 | 0.231 | 0.160 |
| | 336 | 0.588 | 0.270 | 0.175 | 0.335 | 0.259 | 0.172 | 0.399 | 0.262 | 0.366 | 0.279 | 0.245 | 0.169 |
| | 720 | 0.690 | 0.272 | 0.179 | 0.393 | 0.266 | 0.195 | **0.452** | 0.270 | 0.369 | 0.294 | 0.257 | 0.180 |
| **CSDI** (2021) | 96 | 0.823 | 0.244 | 0.161 | 0.711 | 0.227 | 0.160 | 0.811 | 0.250 | 0.365 | 0.593 | 0.229 | 0.163 |
| | 192 | 0.820 | 0.247 | 0.162 | 0.733 | 0.249 | 0.163 | 0.818 | 0.254 | 0.369 | 0.607 | 0.238 | 0.168 |
| | 336 | 0.831 | 0.258 | 0.168 | 0.761 | 0.258 | 0.163 | 0.825 | 0.267 | 0.372 | 0.665 | 0.249 | 0.170 |
| | 720 | 0.838 | 0.254 | 0.170 | 0.792 | 0.266 | 0.170 | 0.833 | 0.281 | 0.380 | 0.692 | 0.245 | 0.184 |
| **TimeGrad** (2021) | 96 | 0.820 | 0.241 | 0.159 | 0.695 | 0.210 | 0.165 | 0.800 | 0.243 | 0.360 | 0.572 | 0.215 | 0.160 |
| | 192 | 0.810 | 0.240 | 0.157 | 0.726 | 0.237 | 0.169 | 0.813 | 0.250 | 0.372 | 0.583 | 0.227 | 0.163 |
| | 336 | 0.830 | 0.251 | **0.142** | 0.735 | 0.240 | 0.170 | 0.815 | 0.253 | 0.365 | 0.597 | 0.239 | 0.179 |
| | 720 | 0.830 | 0.252 | 0.167 | 0.741 | 0.259 | 0.172 | 0.821 | 0.279 | 0.375 | 0.615 | 0.257 | 0.187 |
| **TimeDiff** (2023) | 96 | 0.819 | 0.240 | 0.158 | 0.688 | 0.211 | 0.170 | 0.781 | 0.242 | 0.355 | 0.577 | 0.210 | 0.158 |
| | 192 | 0.805 | 0.236 | 0.150 | 0.702 | 0.224 | 0.173 | 0.805 | 0.246 | 0.370 | 0.580 | 0.229 | 0.163 |
| | 336 | 0.833 | 0.255 | **0.142** | 0.774 | 0.238 | 0.180 | 0.812 | 0.251 | 0.362 | 0.594 | 0.229 | 0.165 |
| | 720 | 0.835 | 0.256 | 0.169 | 0.795 | 0.243 | 0.197 | 0.825 | 0.278 | 0.378 | 0.594 | 0.237 | 0.170 |
| **DTS.** (2024) | 96 | 0.720 | 0.218 | 0.143 | 0.654 | 0.207 | 0.163 | 0.643 | 0.222 | 0.350 | 0.491 | 0.195 | 0.154 |
| | 192 | 0.725 | 0.217 | 0.144 | 0.665 | **0.218** | 0.170 | 0.649 | 0.232 | 0.359 | 0.503 | 0.211 | 0.159 |
| | 336 | 0.722 | 0.220 | 0.145 | 0.674 | **0.225** | 0.170 | 0.647 | **0.225** | 0.350 | 0.519 | 0.218 | 0.160 |
| | 720 | 0.830 | 0.257 | 0.170 | 0.692 | 0.236 | 0.177 | 0.832 | 0.291 | 0.366 | 0.572 | 0.220 | 0.165 |
| **TSFlow** (2025) | 96 | 0.724 | **0.205** | 0.140 | 0.669 | 0.225 | 0.177 | 0.640 | 0.214 | 0.333 | 0.496 | 0.199 | 0.157 |
| | 192 | 0.724 | 0.210 | 0.143 | 0.693 | 0.226 | 0.197 | 0.646 | 0.230 | 0.358 | 0.502 | 0.195 | **0.153** |
| | 336 | 0.725 | 0.223 | 0.148 | 0.705 | 0.227 | 0.190 | 0.657 | 0.232 | 0.358 | 0.539 | 0.207 | 0.155 |
| | 720 | 0.811 | 0.250 | 0.165 | 0.699 | **0.230** | 0.193 | 0.838 | 0.295 | 0.369 | 0.540 | 0.210 | 0.159 |
| **KineFlow** | 96 | 0.383 | 0.207 | **0.135** | 0.307 | **0.195** | 0.154 | **0.311** | 0.210 | **0.319** | 0.269 | **0.183** | 0.152 |
| | 192 | **0.385** | 0.209 | 0.138 | **0.301** | 0.227 | **0.159** | 0.395 | **0.221** | **0.320** | **0.260** | **0.192** | 0.157 |
| | 336 | **0.380** | 0.215 | 0.147 | **0.314** | 0.229 | 0.163 | **0.398** | 0.225 | **0.322** | 0.271 | **0.199** | 0.154 |
| | 720 | **0.387** | 0.219 | 0.140 | 0.337 | 0.235 | **0.166** | 0.482 | 0.227 | **0.329** | 0.280 | **0.200** | **0.157** |

*Table 13.* Full results on benchmark datasets. Comparison of MSE, CRPS, and V-Err on Electricity, Traffic, Weather, and Exchange. **Bold** indicates the best result, and underlined indicates the second best.

| Model | Len | Electricity | | | Traffic | | | Weather | | | Exchange | | |
|---|---|---|---|---|---|---|---|---|---|---|---|---|---|
| | | MSE | CRPS | V-Err | MSE | CRPS | V-Err | MSE | CRPS | V-Err | MSE | CRPS | V-Err |
| **AF.** (2021) | 96 | 0.196 | 0.272 | **0.057** | 0.597 | 0.615 | 0.367 | 0.245 | 0.356 | 0.065 | 0.205 | 0.291 | 0.098 |
| | 192 | 0.212 | 0.291 | 0.066 | 0.609 | 0.630 | 0.371 | 0.323 | 0.363 | 0.081 | 0.315 | 0.305 | 0.112 |
| | 336 | 0.214 | 0.290 | 0.067 | 0.623 | 0.641 | 0.372 | 0.353 | 0.371 | 0.085 | 0.541 | 0.374 | 0.135 |
| | 720 | 0.235 | 0.294 | 0.069 | 0.640 | 0.644 | 0.373 | 0.412 | 0.372 | 0.088 | 1.377 | 0.412 | 0.149 |
| **PTST.** (2023) | 96 | 0.169 | 0.279 | 0.063 | 0.370 | 0.572 | 0.371 | 0.189 | 0.331 | **0.062** | **0.145** | 0.288 | 0.098 |
| | 192 | 0.267 | 0.274 | 0.069 | 0.399 | 0.581 | 0.374 | 0.294 | 0.339 | 0.070 | **0.271** | 0.290 | 0.105 |
| | 336 | 0.263 | 0.280 | 0.072 | 0.492 | 0.589 | 0.376 | 0.345 | 0.274 | 0.077 | 0.492 | 0.356 | 0.115 |
| | 720 | 0.297 | 0.288 | 0.080 | 0.532 | 0.590 | 0.380 | 0.414 | 0.277 | 0.080 | **0.879** | 0.405 | 0.138 |
| **DLinear** (2023) | 96 | 0.240 | 0.278 | 0.063 | 0.470 | 0.575 | 0.374 | 0.276 | 0.332 | 0.065 | 0.149 | 0.287 | 0.114 |
| | 192 | 0.253 | 0.282 | 0.070 | 0.493 | 0.580 | 0.375 | 0.280 | 0.335 | 0.071 | 0.305 | 0.311 | 0.119 |
| | 336 | 0.269 | 0.285 | 0.070 | 0.536 | 0.592 | 0.371 | 0.365 | 0.340 | 0.079 | 0.598 | 0.415 | 0.139 |
| | 720 | 0.273 | 0.289 | 0.078 | 0.566 | 0.590 | 0.370 | 0.383 | 0.351 | 0.084 | 0.911 | 0.472 | 0.148 |
| **iTrans.** (2024) | 96 | 0.149 | 0.280 | 0.069 | 0.390 | 0.571 | 0.370 | 0.175 | 0.336 | 0.070 | 0.198 | 0.288 | 0.105 |
| | 192 | 0.162 | 0.285 | 0.073 | 0.417 | 0.580 | 0.374 | 0.228 | 0.342 | 0.075 | 0.306 | 0.291 | 0.110 |
| | 336 | 0.178 | 0.288 | 0.075 | 0.433 | 0.588 | 0.379 | 0.278 | 0.345 | 0.075 | 0.510 | 0.395 | 0.142 |
| | 720 | 0.225 | 0.288 | 0.079 | 0.467 | 0.600 | 0.381 | 0.358 | 0.360 | 0.091 | 1.247 | 0.399 | 0.146 |
| **TimeXer** (2024b) | 96 | 0.140 | 0.263 | 0.065 | 0.428 | 0.588 | 0.372 | 0.157 | 0.320 | 0.073 | 0.182 | 0.285 | **0.093** |
| | 192 | 0.157 | 0.265 | 0.066 | 0.449 | 0.592 | 0.383 | 0.204 | 0.331 | 0.080 | 0.277 | 0.301 | 0.106 |
| | 336 | **0.176** | 0.265 | 0.066 | 0.471 | 0.597 | 0.383 | 0.261 | 0.332 | 0.089 | 0.435 | 0.372 | 0.139 |
| | 720 | 0.211 | 0.278 | 0.071 | 0.520 | 0.600 | 0.388 | 0.340 | 0.335 | 0.092 | 0.885 | 0.365 | 0.172 |
| **CSDI** (2021) | 96 | 0.337 | 0.262 | 0.067 | 0.915 | 0.560 | 0.370 | 0.450 | 0.319 | 0.074 | 0.331 | 0.287 | 0.095 |
| | 192 | 0.336 | 0.260 | 0.067 | 0.989 | 0.577 | 0.392 | 0.472 | 0.320 | 0.079 | 0.365 | 0.301 | 0.101 |
| | 336 | 0.343 | 0.267 | 0.069 | 1.022 | 0.593 | 0.405 | 0.480 | 0.325 | 0.083 | 0.695 | 0.384 | 0.117 |
| | 720 | 0.355 | 0.282 | 0.073 | 1.022 | 0.596 | 0.407 | 0.486 | 0.329 | 0.085 | 1.453 | 0.411 | 0.130 |
| **TimeGrad** (2021) | 96 | 0.335 | 0.260 | 0.067 | 0.988 | 0.521 | 0.365 | 0.432 | 0.320 | 0.075 | 0.341 | 0.286 | 0.096 |
| | 192 | 0.312 | 0.265 | 0.069 | 0.977 | 0.565 | 0.383 | 0.489 | 0.313 | 0.067 | 0.379 | 0.299 | **0.098** |
| | 336 | 0.340 | 0.260 | 0.065 | 1.015 | 0.590 | 0.399 | 0.472 | 0.313 | 0.080 | 0.690 | 0.392 | 0.110 |
| | 720 | 0.350 | 0.281 | 0.069 | 1.033 | 0.597 | 0.408 | 0.485 | 0.315 | 0.080 | 1.394 | 0.410 | 0.122 |
| **TimeDiff** (2023) | 96 | 0.330 | 0.262 | 0.065 | 0.984 | 0.518 | 0.362 | 0.431 | 0.321 | 0.079 | 0.395 | 0.284 | **0.093** |
| | 192 | 0.302 | 0.261 | 0.067 | 0.972 | 0.564 | 0.380 | 0.488 | 0.310 | 0.065 | 0.402 | 0.289 | 0.100 |
| | 336 | 0.345 | 0.263 | 0.069 | 1.019 | 0.593 | 0.400 | 0.475 | 0.319 | 0.089 | 0.704 | 0.355 | 0.107 |
| | 720 | 0.354 | 0.289 | 0.078 | 1.051 | 0.603 | 0.422 | 0.490 | 0.322 | 0.091 | 1.330 | 0.421 | 0.115 |
| **DTS.** (2024) | 96 | 0.316 | 0.260 | 0.061 | 0.944 | **0.500** | 0.349 | 0.405 | 0.303 | 0.066 | 0.388 | 0.285 | 0.105 |
| | 192 | 0.319 | 0.263 | 0.065 | 0.945 | **0.508** | 0.349 | 0.409 | 0.310 | 0.069 | 0.391 | 0.293 | 0.112 |
| | 336 | 0.319 | 0.260 | 0.062 | 0.948 | 0.505 | 0.371 | 0.405 | 0.308 | 0.070 | 0.655 | 0.342 | 0.114 |
| | 720 | 0.364 | 0.286 | 0.088 | 1.055 | 0.607 | 0.425 | 0.492 | 0.325 | 0.096 | 0.974 | 0.340 | 0.115 |
| **TSFlow** (2025) | 96 | 0.312 | 0.265 | 0.062 | 0.956 | 0.505 | 0.343 | 0.415 | 0.205 | 0.065 | 0.392 | 0.254 | 0.108 |
| | 192 | 0.318 | 0.260 | **0.062** | 0.949 | 0.512 | 0.366 | 0.408 | 0.315 | 0.070 | 0.415 | 0.273 | 0.112 |
| | 336 | 0.322 | 0.265 | 0.066 | 0.949 | **0.492** | 0.370 | 0.401 | 0.308 | 0.077 | 0.736 | **0.319** | 0.112 |
| | 720 | 0.363 | 0.285 | 0.089 | 1.045 | 0.600 | 0.421 | 0.495 | 0.328 | 0.090 | 0.955 | 0.337 | 0.115 |
| **KineFlow** | 96 | **0.110** | **0.257** | 0.059 | **0.298** | 0.506 | **0.342** | **0.102** | **0.172** | 0.063 | 0.185 | **0.243** | **0.095** |
| | 192 | **0.141** | **0.258** | 0.062 | **0.299** | 0.513 | **0.340** | **0.152** | **0.176** | **0.060** | 0.281 | **0.270** | 0.098 |
| | 336 | 0.182 | **0.253** | 0.060 | **0.342** | 0.515 | 0.343 | **0.219** | **0.186** | 0.067 | **0.425** | 0.320 | **0.100** |
| | 720 | **0.207** | **0.255** | 0.060 | **0.388** | 0.523 | 0.345 | **0.270** | **0.189** | 0.065 | 0.900 | **0.335** | **0.105** |

*Table 14.* Full forecasting results on AEMO. Corresponding prediction lengths include $\{12, 24, 48\}$. **Bold** indicates the best result, and underlined indicates the second best.

| | | VIC | | | NSW | | | QLD | | | SA | | | TAS | | |
|---|---|---|---|---|---|---|---|---|---|---|---|---|---|---|---|---|
| | | MSE | CRPS | V-Err | MSE | CRPS | V-Err | MSE | CRPS | V-Err | MSE | CRPS | V-Err | MSE | CRPS | V-Err |
| **AF.** (2021) | 12 | 0.819 | 0.542 | 0.151 | 1.115 | 0.579 | 0.226 | 0.987 | 0.557 | 0.224 | 0.258 | 0.575 | 0.248 | 1.067 | 0.612 | 0.232 |
| | 24 | 1.238 | 1.181 | 0.226 | 1.164 | 0.648 | 0.212 | 1.032 | 0.691 | 0.252 | 0.305 | 0.602 | 0.257 | 1.099 | 0.693 | 0.271 |
| | 48 | 1.206 | 0.632 | 0.208 | 1.243 | 0.619 | 0.209 | 1.318 | 0.705 | 0.253 | 0.351 | 0.595 | 0.280 | 1.157 | 0.702 | 0.285 |
| **PTST.** (2023) | 12 | 1.221 | 0.651 | 0.215 | 1.095 | 0.580 | 0.230 | 1.115 | 0.571 | 0.235 | 0.301 | 0.592 | 0.269 | 0.995 | 0.592 | 0.224 |
| | 24 | 1.215 | 0.648 | 0.219 | 1.155 | 0.590 | 0.241 | 1.210 | 0.565 | 0.231 | 0.295 | 0.584 | 0.264 | 0.991 | 0.599 | 0.235 |
| | 48 | 1.115 | 0.652 | 0.225 | 1.200 | 0.588 | 0.250 | 1.202 | 0.578 | 0.240 | 0.308 | 0.595 | 0.275 | 1.022 | 0.620 | 0.255 |
| **DLinear** (2023) | 12 | 0.663 | 0.522 | 0.188 | 0.995 | 0.653 | 0.241 | 1.014 | 0.593 | 0.283 | 0.227 | 0.583 | 0.211 | 0.976 | 0.605 | 0.231 |
| | 24 | 0.854 | 0.693 | 0.271 | 0.989 | 0.641 | 0.245 | 1.112 | 0.599 | 0.280 | 0.241 | 0.586 | 0.227 | 0.993 | 0.614 | 0.257 |
| | 48 | 0.907 | 0.652 | 0.263 | 1.002 | 0.657 | 0.258 | 1.125 | 0.580 | 0.295 | 0.258 | 0.592 | 0.230 | 0.981 | 0.612 | 0.260 |
| **iTrans.** (2024) | 12 | 0.515 | 0.498 | 0.195 | 0.913 | 0.552 | 0.205 | 0.857 | 0.532 | 0.207 | 0.213 | 0.562 | 0.221 | 0.893 | 0.615 | 0.215 |
| | 24 | 0.712 | 0.701 | 0.251 | 0.892 | 0.613 | 0.227 | 0.963 | 0.559 | 0.208 | 0.227 | 0.570 | 0.229 | 0.917 | 0.621 | 0.223 |
| | 48 | 0.890 | 0.753 | 0.259 | 0.953 | 0.621 | 0.230 | 0.995 | 0.571 | 0.225 | 0.251 | 0.585 | 0.233 | 0.994 | 0.617 | 0.219 |
| **TimeXer** (2024b) | 12 | 0.472 | 0.637 | 0.209 | 0.682 | 0.540 | 0.200 | 0.683 | 0.549 | 0.215 | 0.237 | 0.581 | 0.232 | 0.851 | 0.602 | 0.204 |
| | 24 | 0.653 | 0.910 | 0.218 | 0.690 | 0.592 | 0.209 | 0.602 | 0.552 | 0.219 | 0.240 | 0.588 | 0.235 | 0.881 | 0.615 | 0.215 |
| | 48 | 0.772 | 0.885 | 0.220 | 0.795 | 0.598 | 0.215 | 0.719 | 0.567 | 0.225 | 0.251 | 0.592 | 0.241 | 0.892 | 0.620 | 0.229 |
| **CSDI** (2021) | 12 | 1.222 | 0.513 | 0.194 | 1.257 | 0.503 | 0.192 | 1.001 | 0.499 | 0.202 | 0.692 | 0.512 | 0.207 | 0.996 | 0.557 | 0.201 |
| | 24 | 1.229 | 0.620 | 0.205 | 1.351 | 0.512 | 0.207 | 1.005 | 0.512 | 0.193 | 0.706 | 0.510 | 0.192 | 0.995 | 0.541 | 0.195 |
| | 48 | 1.251 | 0.604 | 0.202 | 1.472 | **0.519** | 0.203 | 1.119 | 0.508 | 0.205 | 0.705 | 0.515 | 0.213 | 1.002 | 0.557 | 0.199 |
| **TimeGrad** (2021) | 12 | 1.127 | 0.505 | 0.221 | 1.133 | 0.509 | 0.190 | 1.005 | 0.501 | 0.198 | 0.689 | 0.505 | 0.192 | 0.992 | 0.543 | 0.189 |
| | 24 | 1.128 | 0.608 | 0.226 | 1.145 | 0.516 | 0.195 | 1.007 | 0.508 | 0.204 | 0.702 | 0.514 | 0.199 | 1.015 | 0.557 | 0.194 |
| | 48 | 1.128 | 0.617 | 0.237 | 1.159 | 0.521 | 0.199 | 1.017 | 0.513 | 0.209 | 0.710 | 0.522 | 0.203 | 1.033 | 0.561 | 0.200 |
| **TimeDiff** (2023) | 12 | 1.215 | 0.492 | 0.219 | 1.185 | 0.498 | 0.197 | 1.013 | 0.510 | 0.194 | 0.690 | 0.513 | 0.190 | 0.990 | 0.535 | 0.192 |
| | 24 | 1.206 | 0.612 | 0.220 | 1.190 | 0.519 | 0.199 | 1.015 | 0.497 | 0.192 | 0.699 | 0.511 | 0.193 | 0.995 | 0.540 | 0.195 |
| | 48 | 1.208 | 0.610 | 0.221 | 1.199 | 0.525 | 0.199 | 1.015 | 0.494 | 0.186 | 0.700 | 0.515 | 0.198 | 0.990 | 0.545 | 0.199 |
| **DTS.** (2024) | 12 | 0.954 | 0.462 | 0.195 | 0.932 | 0.490 | 0.196 | 0.972 | 0.482 | 0.194 | 0.411 | 0.480 | 0.175 | 0.905 | 0.521 | 0.184 |
| | 24 | 0.998 | 0.592 | 0.190 | 0.955 | 0.506 | 0.196 | 0.984 | 0.489 | 0.200 | 0.415 | 0.484 | 0.175 | 0.910 | 0.520 | 0.185 |
| | 48 | 0.990 | 0.590 | 0.192 | 0.998 | 0.544 | 0.195 | 0.990 | 0.491 | 0.205 | 0.422 | 0.488 | 0.176 | 0.913 | 0.533 | 0.187 |
| **TSFlow** (2025) | 12 | 0.877 | 0.459 | 0.186 | 0.901 | 0.495 | 0.190 | 0.966 | 0.472 | 0.190 | 0.395 | 0.472 | 0.176 | 0.911 | 0.513 | 0.180 |
| | 24 | 0.870 | **0.459** | 0.183 | 0.905 | 0.573 | 0.192 | 0.969 | 0.484 | 0.199 | 0.411 | 0.470 | 0.179 | 0.915 | 0.519 | 0.183 |
| | 48 | 0.889 | **0.463** | 0.180 | 0.917 | 0.602 | 0.193 | 0.978 | 0.485 | 0.199 | 0.441 | 0.473 | 0.180 | 0.919 | 0.524 | 0.189 |
| **KineFlow** (Ours) | 12 | **0.330** | **0.442** | **0.128** | **0.418** | **0.466** | **0.145** | **0.418** | **0.439** | **0.151** | **0.117** | **0.327** | **0.114** | **0.410** | **0.447** | **0.143** |
| | 24 | **0.569** | 0.601 | **0.194** | **0.531** | **0.499** | **0.143** | **0.450** | **0.420** | **0.165** | **0.138** | **0.358** | **0.119** | **0.415** | **0.481** | **0.140** |
| | 48 | **0.571** | 0.543 | **0.141** | **0.604** | 0.558 | **0.151** | **0.446** | **0.429** | **0.168** | **0.147** | **0.365** | **0.129** | **0.428** | **0.503** | **0.155** |

