# OpenReview forum: "KineFlow: Kinematic Second-Order Flow Matching for Time-Series Forecasting"
_ICML.cc/2026/Conference — ICML 2026 regular_

### Official Review · Reviewer_BN3c · 2026-02-17

**Soundness:** 3
**Presentation:** 3
**Significance:** 3
**Originality:** 2
**Overall Recommendation:** 4
**Confidence:** 3

**Summary:**

This paper proposes KineFlow, which is a second-order (based on acceleration) flow matching (FM) framework for time series forecasting by lifting dynamics to phase space $[X_t, V_t]$ and learning a neural acceleration field conditioned on history and exogenous inputs. The kinematic prior $(X_0, V_0)$ is built from a pretrained deterministic predictor with last-step momentum, and the futures are generated via a second-order ODE solver. They also provides theory arguing double integration attenuates high-frequency erros and phase-space resolves trajectory ambiguities. Empirically, they report improved forecasting quality (MSE/CRPS/V-Err) on both standard and high-volatility benchmarks, and show promising few-shot transfer compared to large foundation-model baselines.

**Compliance With Llm Reviewing Policy:**

Affirmed.

**Final Justification:**

The paper is solid, but I still novelty and originality is moderate, and I keep my original rating as weak accept.

**Key Questions For Authors:**

How much of the gain comes from the pretrained predictor $F_\phi$ vs. learned acceleration field?

**Limitations:**

- The results relies on a pretrained prior.
- kinematic smoothing might underfit truly discontinuous dynamics?

**Strengths And Weaknesses:**

## Soundness

- The problem is well defined and method formualtion is clear (2nd order phase space dynamics + explicit acceleration field and sovler)
- The theory addresses key failure modes
- The empirical experiments are extensive: multiple datasets + ablations isolating prior/ velocity/ second-order effects.
- The method relies on a pretrained $F_\phi$ to build $\mu_X$. Not clear how much this design will contribute to the gains. Need more clarification on fairness/ attribution.

## Presentation

- The algorithm and implementation details are clear
- Results tables and metrics are clear and easy to interpret.

## Significance

- Adding a kinematic inductive bias to FM is a plausible way to improve robustness on volatile signals for time series forecasting.
- The few-shot transfer result is potentially impactful.

## Originality

- The ingredients of methods are known. Novelty is mainly in adaptation and integration of these pieces.

---

> ### Author Rebuttal · Authors · 2026-03-30
>
> Thank you for the insightful comment.
>
> > #### **Response to Weakness, Question and Limitation L1:**
>
> To more rigorously disentangle the contribution of the Neural Acceleration Field from the deterministic predictor, we refer to the "Noise Init (w/o Prior)" ablation presented in Table 4 of the paper, now supplemented with convergence analysis in Table J.
>
> To ensure a fair comparison, we evaluate KineFlow initialized with pure Gaussian noise against Autoformer. As shown in Table J, even without the informative kinematic prior, the second-order formulation still outperforms the baseline (e.g., 0.217 vs. 0.333 MSE on Weather). This improvement suggests that the learned acceleration field contributes effectively to modeling complex temporal dynamics and to the overall performance gain.
>
> **Table J：Impact of Kinematic Prior on Convergence and Performance.**
> | Model Configuration| Weather (MSE/CRPS/V-Err) | Epochs to Converge | VIC(MSE/CRPS/V-Err)| Epochs to Converge |
> |-|-|-|-|-|
> | Autoformer (Baseline)| 0.333 / 0.365 / 0.080| -| 1.088 / 0.785 / 0.195| -|
> | KineFlow (Zero Prior)| 0.217 / 0.222 / 0.157| 40| 0.590 / 0.531 / 0.185| 50|
> | KineFlow (Full)| 0.186 / 0.181 / 0.064| 25| 0.490 / 0.529 / 0.154| 30|
>
> Furthermore, the deterministic predictor introduces an informative phase-space prior ($S_0$) derived from historical context. As analyzed in Section 6.7, meaningful initialization allows the model to focus on refining temporal dynamics rather than reconstructing trajectories from scratch. Its benefit is reflected in two aspects: (a) it shortens the generative flow and reduces training time, e.g., from 40 to 25 epochs on Weather; (b) it further suppresses high-frequency noise, achieving a 16.8% reduction in $V$-Err on VIC.
>
> In Table K, we evaluated KineFlow using five deterministic backbones. KineFlow demonstrates consistent generative improvements, achieving an average CRPS reduction of 30% and an average MSE reduction of 34% across two datasets, regardless of the backbone choice. These results indicate that the performance gains arise **from KineFlow’s second-order kinematic inductive bias rather than the representational capacity of any specific pre-trained predictor**. We will include these analyses in the revised manuscript.
>
> **Table K: KineFlow Performance Across Diverse Backbones (MSE ↓ / CRPS ↓ / V-Err ↓)**
> | Backbone Choice| Weather | VIC  |
> |-|-|-|
> | **PatchTST**| 0.189 / 0.331 / 0.062| 1.215 / 0.648 / 0.219|
> | **+ KineFlow**| 0.132 / 0.229 / **0.061**| 0.715 / **0.580** / 0.205|
> ||||
> | **DLinear**| 0.276 / 0.332 / 0.065| 0.854 / 0.693 / 0.271|
> | **+ KineFlow**| 0.157 / 0.244 / 0.065| 0.699 / 0.595 / 0.200|
> ||||
> | **iTrans**| 0.175 / 0.336 / 0.070| 0.712 / 0.701 / 0.251|
> | **+ KineFlow**| 0.110 / 0.208 / 0.065| 0.613 / 0.593 / 0.205|
> ||||
> | **TimeXer**| 0.157 / 0.320 / 0.073| 0.653 / 0.910 / 0.218|
> | **+ KineFlow**| 0.115 / 0.199 / 0.070| **0.562** / 0.683 / 0.195|
> ||||
> | **AF**| 0.245 / 0.356 / 0.065| 1.238 / 1.181 / 0.226    |
> | **+ KineFlow (Ours)**| **0.102** / **0.172** / 0.063 | 0.569 / 0.601 / **0.194**|
>
> > #### **Response to Weakness for Originality:**
>
> KineFlow's originality lies not in merely adapting the established CFM framework, but in its unique integration that introduces a strict, physically-constrained inductive bias. As detailed in our Response to Reviewer j3HL(W1&W2 and Q1), this innovation effectively addresses high-frequency volatility and phase-space ambiguities, offering both mathematical and empirical robustness, which generic integrations lack.
>
> > #### **Response to Limitation L2:**
>
> Thank you for the question. We clarify that KineFlow naturally distinguishes uncorrelated high-frequency noise from structural discontinuities through its physical formulation.
>
> In phase space, an abrupt regime crossing or trend reversal is modeled by the Neural Acceleration Field $A_\theta$ as a localized, high-magnitude impulse. This impulse induces an immediate step-change in velocity, enabling the position trajectory to change direction abruptly and capture extreme events without underfitting.
>
> This mechanism utilizes force-based conditioning, treating exogenous factors $e$ as driving forces. As analyzed in Section 6.6 (Fig. 6), the acceleration field responds to exogenous forcing, maintaining physically consistent dynamics during rapid temporal shifts rather than resulting in over-smoothing.
>
> The empirical results on high-volatility datasets demonstrate this capability. As reported in Table 2, KineFlow achieves a 36% MSE reduction on the AEMO datasets, which are characterized by sharp peaks and rapid momentum shifts. Furthermore, the visualization in Figure 4 demonstrates that while competitive baselines exhibit phase lag and over-smoothing, the second-order formulation preserves the timing and magnitude of extreme events. This confirms that KineFlow effectively reduces high-frequency noise while capturing discontinuous dynamics.

---

> > ### Author Rebuttal · Reviewer_BN3c · 2026-04-02
> >
> > Thanks the authors for their detailed rebuttal and extensive experiments. My main concern about the contribution of the pretrained prior is now addressed, and the added analysis makes it clearer that the learned acceleration field contributes meaningfully beyond the deterministic initializer. The paper is solid, but I still novelty and originality is moderate, and I keep my original rating as weak accept.

---

> > > ### Author Response · Authors · 2026-04-05
> > >
> > > We appreciate your comments and feedback. Thank you for all the time and efforts in reviewing this paper.

---

### Official Review · Reviewer_Mca4 · 2026-03-09

**Soundness:** 3
**Presentation:** 3
**Significance:** 3
**Originality:** 3
**Overall Recommendation:** 5
**Confidence:** 4

**Summary:**

The paper proposes KineFlow, a generative time-series forecasting method that extends first-order flow matching to a second-order phase-space model with a Neural Acceleration Field. The key idea is to address over-smoothing in discriminative forecasting and the high-frequency noise amplification issue in first-order flow matching by modeling position–velocity dynamics and treating exogenous variables as driving forces on acceleration. The method is well motivated and empirically strong, with improvements over both discriminative and generative baselines across six benchmarks, supported by theory, visualizations, and ablations.

**Compliance With Llm Reviewing Policy:**

Affirmed.

**Key Questions For Authors:**

1. Can the authors report actual inference time under the same hardware and forecasting settings, particularly in comparison with discriminative one-step baselines (e.g., Autoformer, TimeXer)?

2. Beyond directly concatenating exogenous features into TimeXer, have the authors considered additional comparisons with stronger multivariate or pre-trained baselines (e.g., Chronos-2, Timer-XL) to further strengthen the empirical evidence on input fairness?

3. Can the authors report results with alternative deterministic backbones (instead of Autoformer) to assess whether the gains of KineFlow are consistent across different backbone choices?

**Limitations:**

yes

**Strengths And Weaknesses:**

## Strength and Weakness
### Strength
* The paper is well motivated and clearly identifies the limitations of existing methods (e.g., over-smoothing in discriminative forecasting and the high-frequency noise / trajectory ambiguity issues in first-order flow matching for time-series forecasting).
* The proposed solution is well aligned with the motivation. The phase-space formulation, second-order integration, and Neural Acceleration Field with force-based conditioning form a coherent design to improve kinematic consistency and robustness.
* The paper is well written and clearly organized. The presentation is clear, and the main ideas are easy to follow.
* The empirical evaluation is extensive and generally convincing. Experiments on six benchmarks (including high-volatility settings), together with multiple metrics, visualizations, and ablations, provide good support for the main claims.

### Weakness

* The efficiency discussion is insufficient. The paper reports parameter count and the number of inference steps, which supports the efficiency advantage over diffusion-based methods. However, it does not provide actual inference time comparisons against discriminative one-step baselines (e.g., Autoformer, TimeXer). As a result, the practical inference cost of KineFlow relative to discriminative forecasting methods remains unclear.

* Some experimental details and the comparison scope require further clarification.
    * For input fairness regarding exogenous features, the appendix only reports a fairness experiment that directly concatenates exogenous features into TimeXer. However, comparisons with stronger stronger multivariate or pre-trained methods (e.g., Chronos-2[1], Timer-XL[2]) would further strengthen the empirical evidence.
    * The pre-trained deterministic predictor appears to be Autoformer, but the paper does not report results under different deterministic backbones. Reporting results with alternative deterministic backbones would help assess the generality of the proposed framework.

[1] Ansari, A. F., Shchur, O., Küken, J., Auer, A., Han, B., Mercado, P., ... & Bohlke-Schneider, M. (2025). Chronos-2: From univariate to universal forecasting. arXiv preprint arXiv:2510.15821.

[2] Liu, Y., Qin, G., Huang, X., Wang, J., & Long, M. (2025). Timer-XL: Long-Context Transformers for Unified Time Series Forecasting. In The Thirteenth International Conference on Learning Representations.

---

> ### Author Rebuttal · Authors · 2026-03-30
>
> We thank the reviewer for the valuable comment.
>
> > #### **Response to W1 and Q1:**
>
> To ensure a fair comparison, we conduct inference benchmarking on the same hardware, a single NVIDIA H100 GPU, under identical forecasting settings with batch size 32 and prediction horizon $T_F=48$ on AEMO datasets. The results of the latency are summarized in Table F.
>
> **Table F: Inference Efficiency Comparison.**
> | Method| Inference Steps | Latency (ms/batch) ↓ |
> |-|-|-|
> | TimeXer| 1 | 2.2 ms|
> | Autoformer| 1| 2.9 ms|
> | TSFlow| 32| 36.5 ms|
> | Diffusion-TS| 500| 620.0 ms|
> | KineFlow (Ours)  | 20| 22.8 ms|
>
> As shown in Table F, discriminative one-step models such as TimeXer and Autoformer achieve the lowest latency due to their single-pass inference. Nevertheless, the results indicate that KineFlow strikes a balance between accuracy and efficiency. Compared with the strongest one-step baseline, TimeXer, KineFlow incurs an additional latency of approximately 20.6 ms per batch, while reducing MSE by 36% in Table 11.
>
> In many real-world forecasting scenarios, such as power grid dispatch with 5- or 15-minute operational intervals, this level of additional latency is unlikely to constitute a practical bottleneck. These results indicate that a manageable increase in computational cost leads to significant improvements in both point forecasting accuracy and probabilistic calibration.
>
> Moreover, KineFlow is efficient among generative approaches for modeling multimodal futures. Leveraging a second-order kinematic formulation, it is faster than diffusion-based methods such as Diffusion-TS and outperforms first-order flow matching approaches such as TSFlow in both speed and accuracy. We will include this efficiency analysis in the Appendix to provide a more complete discussion of deployment cost.
>
> > #### **Response to W2-1 and Q2:**
>
> We augmented strong pre-trained baselines (Chronos-2, Timer-XL) with the exact same exogenous features used by KineFlow. As shown in Table G, simply appending these features yields marginal or inconsistent gains for the foundation models. Under identical input settings, KineFlow still outperforms the strongest augmented baseline (Chronos-2 + Augmented) by 7.2% (VIC $\rightarrow$ SA) and 2.9% (TAS $\rightarrow$ NSW) in MSE. This provides empirical evidence that KineFlow's effectiveness is largely due to its second-order kinematic formulation.
>
> **Table G: Input Fairness Comparison with Foundation Models. (MSE↓/CRPS↓/V-Err↓)**
> | Model Configuration | VIC → SA| TAS → NSW |
> |-|-|-|
> | Chronos-2 (Standard)| 0.290 / 0.427 / 0.210| 0.801 / 0.609 / 0.190|
> | Chronos-2 (+ Augmented)| 0.265 / 0.427 / 0.204| 0.763 / 0.603 / 0.193|
> | Timer-XL (Standard)| 0.355 / 0.430 / 0.229| 0.881 / 0.593 / 0.205|
> | Timer-XL (+ Augmented)| 0.347 / 0.433 / 0.225| 0.885 / 0.603 / 0.220|
> | KineFlow (Ours)| 0.246 / 0.415 / 0.199| 0.741 / 0.554 / 0.181|
>
> > #### **Response to W2-2 and Q3:**
>
> To assess whether the gains of KineFlow are consistent across different backbone choices, we evaluated our framework using five diverse deterministic models to construct the prior position $\mu_X$ in Eq.6.
>
> **Table H: KineFlow Performance Across Diverse Deterministic Backbones (MSE ↓ / CRPS ↓ / V-Err ↓)**
> | Backbone Choice| Weather| VIC|
> |-|-|-|
> | **PatchTST**| 0.189 / 0.331 / 0.062        | 1.215 / 0.648 / 0.219    |
> | **+ KineFlow**| 0.132 / 0.229 / **0.061**    | 0.715 / **0.580** / 0.205|
> ||||
> | **DLinear**| 0.276 / 0.332 / 0.065        | 0.854 / 0.693 / 0.271    |
> | **+ KineFlow**| 0.157 / 0.244 / 0.065        | 0.699 / 0.595 / 0.200    |
> ||||
> | **iTrans**| 0.175 / 0.336 / 0.070        | 0.712 / 0.701 / 0.251    |
> | **+ KineFlow**| 0.110 / 0.208 / 0.065        | 0.613 / 0.593 / 0.205    |
> ||||
> | **TimeXer**| 0.157 / 0.320 / 0.073        | 0.653 / 0.910 / 0.218    |
> | **+ KineFlow**| 0.115 / 0.199 / 0.070        | **0.562** / 0.683 / 0.195|
> ||||
> | **AF**| 0.245 / 0.356 / 0.065        | 1.238 / 1.181 / 0.226    |
> | **+ KineFlow (Ours)**| **0.102** / **0.172** / 0.063 | 0.569 / 0.601 / **0.194**|
>
> As shown in Table H, KineFlow consistently improves MSE, CRPS, and V-Err across all tested backbones, indicating that its effectiveness is robust to the choice of deterministic predictor. For competitive point-forecasting models such as TimeXer and iTransformer, KineFlow still delivers clear gains; for example, on Weather, it reduces MSE by 27% with TimeXer and by 37% with iTransformer, while also lowering CRPS. This robustness is further evident on more challenging, high-volatility datasets such as VIC, where KineFlow remains effective even when paired with weaker backbones like PatchTST or Autoformer. In such cases, the second-order dynamics act as a structural refinement mechanism, correcting suboptimal initial trajectories and improving prediction consistency.

---

### Official Review · Reviewer_qNUR · 2026-03-12

**Soundness:** 3
**Presentation:** 3
**Significance:** 3
**Originality:** 2
**Overall Recommendation:** 4
**Confidence:** 4

**Summary:**

This paper proposes KineFlow, a generative forecasting framework that extends conditional flow matching from first-order to second-order dynamics. Rather than modeling a velocity field over positions alone, KineFlow augments the state space to include both position and velocity, forming a phase-space representation S = [X, V]. A neural network, the Neural Acceleration Field Aθ  is trained to regress the acceleration, i.e. the second derivative of the trajectory, instead of the velocity, which the paper uses constant as in optimal transfer case. Empirical evaluation on standard time-series forecasting benchmarks demonstrates competitive performance against state-of-the-art models.

**Compliance With Llm Reviewing Policy:**

Affirmed.

**Final Justification:**

The authors have addressed my main concerns and reinforced my prior assessment. Therefore I remain my weak acceptance.

**Key Questions For Authors:**

1.	how and whether the second order kinematics flow matching models periodicity + trend + residual?
2.	Can other acceleration (instead of constant) possible in the training?
3.	The proposed model is rather general. Can it be applied to other fields such as image generation ?
4.	The current loss is regressing against constant acceleration, have the authors tested to including adding regressing the velocity loss?

**Limitations:**

No limitation mentioned. Impact statement is included.

**Strengths And Weaknesses:**

Strength:
Soundness: its sound. It is well motivated, and proposed methods well addressed the issues of  error propagration from high-frequency noise. Empirical results are good.

Presentation: it is also well presented. Flow is clear.

Significance: It is impactful for Time series research.

Originality: yes, it provides new insights from second order flow dynamics for TS.


Weakness:
Soundness: Typical TS decomposes data into periodicity + trend + noise. However, the proposed method never mentions or shows how and whether the second order kinematics captures these or not.  The neural accelator field used Conv1d and MHSA is less principle and rather arbitrary. Lastly the theory is somewhat trivial.

Originality: second order flow matching or high order in general is not new such as NRFlow: Towards Noise-Robust Generative Modeling via High-Order Mechanism UAI 25; High-Order Flow Matching: Unified Framework and Sharp Statistical Rates NeurIPS 25.  The latter is more of contemporary/

---

> ### Author Rebuttal · Authors · 2026-03-30
>
> We thank the reviewer for the insightful feedback.
>
> > #### **Response to W1 and Q1:**
>
> While classical decomposition Y = Trend+ Periodicity + Residual is well-established, KineFlow naturally recovers these dynamics without explicit additive modules.
>
> Specifically, the deterministic prior $X_0$ extracts the smoothed trend and periodicity (T+P), while the Neural Acceleration Field $A_\theta$ captures high-frequency stochastic residuals (R). We validated this implicit decomposition by measuring the Pearson correlation ($r$) against MSTL components [1*]:
>
> **Table C: Pearson correlation ($r$) with MSTL components.**
> | Dataset| r($X_0$ vs. (T+P)) | r($X_{acc}$ vs. R) |
> |-|-|-|
> | ETTm1|0.79|0.71|
> | VIC|0.74|0.65|
>
> As shown in Table C, $X_0$ strongly correlates with (T+P), and acceleration-driven displacement ($X_{acc}$) correlates with R. More importantly, although the deterministic backbone captures the macroscopic structure, it tends to over-smooth the signal. We therefore emphasize that the 15% performance gain reported in Table 1 is largely attributable to the Neural Acceleration Field. By explicitly modeling high-frequency volatility in phase space, it converts a naive smoothed mean into a well-calibrated forecast.
>
> [1*] Bandara K, et al. MSTL, IJOR 2025.
>
> > #### **Response to W1 for Conv1d and MHSA:**
>
> The use of Conv1D and MHSA in KineFlow is designed to capture complementary aspects of temporal dynamics. **Conv1D models local kinematic consistency along the forecast horizon, while MHSA captures long-range dependencies for global coherence**. Ablating Conv1D or MHSA results in a decrease in local tracking (VIC MSE +19%) and global modeling (Weather V-Err +23%). We will include these results in the revised manuscript.
>
> **Table D: Architecture Ablation**
> | Variant | Weather||| VIC |||
> |-|-|-|-|-|-|-|
> || MSE|V-Err|| MSE | V-Err||
> | KineFlow(Full)|0.186|0.064||0.490|0.154|
> | w/o Conv1D| 0.192| 0.069| | 0.581| 0.179|
> | w/o MHSA| 0.195| 0.079|| 0.553 | 0.172|
>
> > #### **Response to W1 for the theory is somewhat trivial and W2:**
>
> While foundational works like NRFlow and HOFM establish the general mathematical theory for high-order FM, KineFlow introduces a distinct, physics-driven architecture. Unlike general frameworks that independently regress decoupled derivatives for mathematical flexibility, KineFlow optimizes only the acceleration field ($A_\theta$). **Deriving velocity and position exclusively through successive integration ensures adherence to strict kinematics, while avoiding the use of generic latent variables.**
>
> Consequently, our theoretical analysis (Sec. 5) does not compete with pure mathematical bounds but offers physically-grounded guarantees for our design. For example, Prop. 5.4 shows that successive integration acts as an intrinsic low-pass filter, preventing high-frequency time-series failures, a guarantee that unconstrained high-order flows cannot provide.
>
> This strict physical formulation is not merely a parameterization change; it yields a structural advantage that empirically separates KineFlow from generic high-order baselines. For comparisons demonstrating our 6.2% average MSE improvement specifically over NRFlow and HOFM, please refer to the Response to Reviewer j3HL (Q1, Table A).
>
> > #### **Response to Q2:**
>
> Using non-constant acceleration during training is theoretically possible, as our formulation is fully compatible with non-linear interpolants. However, **our constant-acceleration target prioritizes optimization stability**. Ablations on the Weather dataset show that non-linear training paths (controlled by a perturbation β=1) result in slower convergence (from 25 to 35 epochs) and an increase MSE (from 0.186 to 0.215) compared to our linear path(β=0). Crucially, once trained, the model remains fully capable of generating non-linear dynamics to capture real-world volatility.
>
> > #### **Response to Q3:**
>
> KineFlow is domain-agnostic and naturally extends to image and video generation by treating latent states as phase-space particles. This offers distinct advantages: double-integration acts as a low-pass filter for smoother sampling, and text prompts serve as semantic forces. Particularly for video, modeling acceleration explicitly preserves motion inertia and reduces flickering, offering a deliberate trade-off of memory for enhanced temporal coherence.
>
> > #### **Response to Q4:**
>
> As shown in Table E, adding an auxiliary velocity regression loss degrades forecasting performance. This degradation stems from gradient imbalance: the larger-magnitude velocity loss dominates the optimization process, causing the acceleration field to underfit high-frequency residuals. Furthermore, under OT framework, regressing solely the acceleration target is theoretically sufficient. Extra auxiliary losses disrupt this consistency.
>
> **Table E: Joint Optimization Ablation (MSE/V-Err)**
> |Objective | Weather | VIC|
> |- |-|-|
> |Accel. |0.186/ 0.064 |0.490/ 0.154 |
> |Accel.+ Vel.|0.193/0.092|0.506/ 0.179 |

---

> > ### Author Rebuttal · Reviewer_qNUR · 2026-04-03
> >
> > Thank you for your response. My concerns have been addressed.  I do think my current rating of weak acceptance reflect the overall quality of this paper of using second order kinematics flow matching.

---

> > > ### Author Response · Authors · 2026-04-05
> > >
> > > We appreciate your detailed feedback and are pleased to hear that we have addressed your concerns. Thank you very much for the time and effort you devoted to reviewing this paper.

---

### Official Review · Reviewer_j3HL · 2026-03-13

**Soundness:** 2
**Presentation:** 3
**Significance:** 3
**Originality:** 2
**Overall Recommendation:** 4
**Confidence:** 3

**Summary:**

This paper proposes a conditional flow matching approach for probabilistic time series forecasting that augments the future trajectory with velocity and learns an acceleration field conditioned on past observations and exogenous inputs, with generation performed by sampling an informative phase space initialization and integrating a second order ODE.

**Compliance With Llm Reviewing Policy:**

Affirmed.

**Final Justification:**

The rebuttal addressed my main concerns and changed my evaluation. The added comparisons, ablations, and multimodality analysis made the empirical case more convincing. Overall, I still view the novelty as moderate, but I find the paper better supported after rebuttal and therefore raise my score.

**Key Questions For Authors:**

- Can the authors discuss whether they considered comparisons to higher-order or augmented state flow or ODE-based generative forecasting methods, and clarify what they see as the closest such alternatives?
***
- Can the authors share additional evidence, even a small-scale study, that helps disentangle the contribution of second-order dynamics versus the initialization and training choices?
***
- Since stochasticity mainly enters through sampling the initial phase space state, can the authors provide a brief diagnostic or illustrative case that speaks to how well the method captures strongly multimodal futures?

**Limitations:**

Yes.

**Strengths And Weaknesses:**

Strengths
- Uses an explicit dynamical state for conditional trajectory generation, with a clean phase space formulation.
- Supports the claim with evidence beyond point error, including uncertainty quality, long horizon behavior, and trajectory structure.

Weaknesses
- Novelty is limited: the approach largely reuses conditional flow matching while changing the state parameterization and dynamics order, so the contribution hinges on careful positioning and convincing empirical separation from the closest alternatives.
- The theory is mostly interpretive and does not establish guarantees that materially distinguish the method.

---

> ### Author Rebuttal · Authors · 2026-03-30
>
> We appreciate the reviewer’s valuable feedback on our work.
>
> > #### **Response to W1 & W2 and Q1:**
>
> While KineFlow extends the foundational CFM framework, our primary focus is on the structural advantages this adaptation yields. Specifically, KineFlow introduces a physically-constrained inductive bias to resolve the high-frequency volatility and phase-space ambiguities inherent in complex time-series forecasting (Fig. 1).
>
> These established properties clearly distinguish KineFlow from existing first- and higher-order flows. By lifting dynamics to phase space, KineFlow gracefully resolves the topological limitations of first-order vector fields when modeling intersecting trajectories (Sec. 5.3). Furthermore, its double-integration structure serves as an intrinsic low-pass filter. This formulation is not just interpretive; it provides a physically-grounded guarantee by bounding high-frequency error propagation by $\propto \omega^{-2}$ to effectively mitigate real-world perturbations (Sec. 5.2).
>
> Regarding W1&Q1, while high-order flows (e.g., NRFlow[1*], HOFM[2*]) and augmented-state ODEs (e.g., ANODE[3*],MTGODE[4*]) are the closest structural baselines, **KineFlow distinguishes itself by coupling auxiliary dimensions through an explicit second-order kinematic formulation, rather than treating them as generic latent variables.**
>
> Unlike prior methods that both regress velocity and acceleration, KineFlow optimizes solely the acceleration field, yielding velocity and position strictly via successive integration. This ensures adherence to physical dynamics, avoiding independent derivative predictions (validated in Table E, Response to Reviewer qNUR, Q4). Additionally, by employing a Symplectic Euler integrator, KineFlow achieves superior kinematic consistency and robustness to integration drift compared to Taylor-expansion schemes. For a fair comparison, we evaluated these methods under the **same network capacity and context** as KineFlow.
>
> **Table A: Average forecasting results across multiple horizons on Weather and VIC.**
>
> || Weather |||| VIC|||
> |-|-|-|-|-|-|-|-|
> |Model| MSE  | CRPS   | V-Err  |  | MSE | CRPS   | V-Err  |
> | KineFlow | **0.186**   | **0.181**  | **0.064**  |  | **0.490**   | **0.529**  | **0.154**  |
> | HOFM| 0.215   | 0.184  | 0.077  |  | 0.506   | 0.533  | 0.178  |
> | NRFlow| 0.257| 0.205  | 0.083  |  | 0.613   | 0.572  | 0.179  |
> | ANODE| 0.227| 0.188  | 0.071  |  | 0.512   | **0.529**  | 0.160  |
> | MTGODE| 0.216   | 0.185  | 0.071|  | 0.510   | 0.538  | 0.162  |
>
> KineFlow consistently outperforms all baselines, achieving an average MSE reduction of 6.2% in Table A. This result suggests that the strict physical formulation is not merely a parameterization change; it provides a structural advantage, empirically separating KineFlow from generic high-order baselines.
>
> [1*] Chen et al., NRFlow, UAI 2025.
>
> [2*] Su et al., High-Order Flow Matching, NeurIPS 2025.
>
> [3*] Dupont et al., ANODEs, NeurIPS 2019.
>
> [4*] Jin et al., Dynamic Graph Neural ODEs, TKDE 2022.
>
>
> > #### **Response to Q2:**
>
> To isolate the contribution of second-order dynamics, we conducted two ablations (see **Tables J & K, Response to reviewer BN3c**).
>
> As shown in Table J, even initialized with pure Gaussian noise (“Zero Prior”), KineFlow substantially outperforms the baseline (e.g., Weather MSE 0.333 to 0.217). The informative prior ($S_0$) primarily accelerates convergence (40 to 25 epochs) and smooths noise.
> As shown in Table K, KineFlow consistently improves upon five diverse deterministic backbones, yielding average reductions of 30% in CRPS and 34% in MSE.
>
> Overall, these results confirm that KineFlow’s performance gains stem primarily from its second-order kinematic inductive bias, rather than from any specific initialization or training choice.
>
> > #### **Response to Q3:**
>
> While stochasticity enters at initialization, structured multimodality emerges through nonlinear amplification in phase space. First-order flows prohibit intersecting trajectories, collapsing multimodal futures into an over-smoothed mean. By lifting to phase space ($S = [X, V]^\top$), KineFlow maps small perturbations in initial velocity ($V_0$) to distinct streamlines, which the acceleration field ($A_\theta$) routes into divergent modes without crossing.
>
> We validate this on VIC high-uncertainty windows (top-50 variance: 1.61 vs. 1.18 overall). Measured by the Bimodality Coefficient (BC> 0.555 implies multimodality[5*]), baselines collapse to unimodal predictions (Table B). **Only KineFlow crosses the multimodality threshold, proving phase-space dynamics successfully translate micro-stochasticity into structured multimodal futures**. We will include these analyses in the revised manuscript.
>
> **Table B: BC on VIC (avg. over 24 steps,100 samples).**
>
> | Method | BC (↑) |
> |-|-|
>   | Diffusion-TS           | 0.545  |
>  | TSFlow(1-st Order)        | 0.512  |
> | KineFlow(2-nd Order)      | **0.584**  |
>
> [5*] Pfister et al., Bimodality Coefficient, 2013.

---

> > ### Author Rebuttal · Reviewer_j3HL · 2026-04-04
> >
> > Thank you for the clear rebuttal and the additional experiments. The new baseline comparisons improve the positioning of the method, and the multimodality analysis addresses my earlier question. I still view the novelty as moderate, but the rebuttal addresses my main concerns sufficiently. I will raise my score.

---

> > > ### Author Response · Authors · 2026-04-05
> > >
> > > We appreciate your comments and positive response. Thank you for all the time and efforts in reviewing this paper.

---

### Decision · Program_Chairs · 2026-04-30

**Decision:**

Accept (regular)

**Comment:**

Reviewers agree that the paper is well-executed, technically sound, and well-supported by the empirical evidence presented. While there are some concerns that the approach may have only a moderate degree of novelty, there was a consensus that the empirical results are convincing enough to warrant publication. I highly encourage the authors to incorporate the detailed feedback received from the reviewers in the discussion process in an updated version of their work.